# Building programmable multicompartment artificial cells incorporating remotely activated protein channels using microfluidics and acoustic levitation

Jin Li [1✉], William D. Jamieson [2], Pantelitsa Dimitriou [1], Wen Xu [3], Paul Rohde [4], Boris Martinac [4,5], Matthew Baker [6], Bruce W. Drinkwater [7✉], Oliver K. Castell [2✉] & David A. Barrow[1✉]

Intracellular compartments are functional units that support the metabolism within living cells, through spatiotemporal regulation of chemical reactions and biological processes. Consequently, as a step forward in the bottom-up creation of artificial cells, building analogous intracellular architectures is essential for the expansion of cell-mimicking functionality. Herein, we report the development of a droplet laboratory platform to engineer complex emulsion-based, multicompartment artificial cells, using microfluidics and acoustic levitation. Such levitated models provide free-standing, dynamic, definable droplet networks for the compartmentalisation of chemical species. Equally, they can be remotely operated with pneumatic, heating, and magnetic elements for post-processing, including the incorporation of membrane proteins; alpha-hemolysin; and mechanosensitive channel of large-conductance. The assembly of droplet networks is three-dimensionally patterned with fluidic input configurations determining droplet contents and connectivity, whilst acoustic manipulation can be harnessed to reconfigure the droplet network in situ. The mechanosensitive channel can be repeatedly activated and deactivated in the levitated artificial cell by the application of acoustic and magnetic fields to modulate membrane tension on demand. This offers possibilities beyond one-time chemically mediated activation to provide repeated, non-contact, control of membrane protein function. Collectively, this expands our growing capability to program and operate increasingly sophisticated artificial cells as life-like materials.

[1] School of Engineering, Cardiff University, The Parade, Cardiff CF24 3AA, UK. [2] School of Pharmacy and Pharmaceutical Sciences, Cardiff University, King Edward VII Ave, Cardiff CF10 3NB, UK. [3] Cardiff Business School, Cardiff University, Aberconway Building, Colum Dr, Cardiff CF10 3EU, UK. [4] Victor Chang Cardiac Research Institute, Lowy Packer Building, 405 Liverpool St, Darlinhurst, NSW 2010, Australia. [5] School of Clinical Medicine, UNSW, Sydney, NSW 2052, Australia. [6] School of Biotechnology and Biomolecular Science, UNSW, Sydney, NSW 2052, Australia. [7] Department of Mechanical Engineering, University of Bristol, University Walk, Bristol BS8 1TR, UK. ✉email: LiJ40@cardiff.ac.uk; B.Drinkwater@bristol.ac.uk; CastellO@cardiff.ac.uk; Barrow@cardiff.ac.uk

Through evolution, eukaryotic cells have exploited intra-cellular compartmentalisation to increase the efficiency and range of functional complexity in cellular metabolism[1]. There is wide interest in harnessing such principles in the creation of life-like synthetic materials. The bottom-up construction of artificial cells aims to harness and imitate some of these key cellular functions, with de novo structures[2,3]. Protocells mimic possible primitive cells, and have been routinely fabricated by forming small, singularly compartmentalised droplets or vesicles with relatively simple infrastructure[4–6], usually built through the macromolecular assembly of lipids or other amphiphiles[7,8]. Such basic models can encapsulate different reagents, representing minimal 'cellular' systems for cell-free studies, including those of membrane properties[9], chemically mediated communication[10–12], and the manipulation of genetic information[13]. State-of-art protocell materials can interact with living cells[14] for potential biotechnological applications, such as immunogenicity enhancement[15], organoid formation[16] or blood vessel vasodilation[17].

To fabricate increasingly complex artificial cells, a key objective is to develop functionalities that are underpinned by intracellular compartmentalised architectures. Such structures can be formed by the encapsulation of lipid-bounded aqueous droplets and biomolecule complexes, within a host (envelope) droplet[18,19], where each compartment may contain different biochemical species. Recent work has demonstrated that organelle-like components can work as functional units to process molecular signals[20,21], regulate sequential reactions[22–24], and may be used for energy harvesting[25–28] within artificial cells. Furthermore, compartmentalisation has been harnessed in similar ways in the packing and patterning of individual protocells to construct tissue-like materials, incorporating protein channels[29], DNA sequences[30], and functional hydrogels[31]. These works indicate that structural complexity can display a range of emergent properties defined by the contents and connectivity of constituent components in such soft, biomimetic materials. However, few efforts have focused on the permutation of compartments within artificial cells, and the subsequent processing and functionality of these architectures. This is, in part, because high-order emulsification processes for the creation of multi-compartmental structures with controllable (bio) chemical reagent distribution, remain a significant challenge. Most work to date has relied largely on the manual juxtaposition, or 3D printing, of multiple droplets to create tissue-like materials[32–34]. Droplet microfluidics provides the ability to control emulsion formation and has become an invaluable tool in the fabrication of vesicles and protocells[35–37]. Meanwhile, acoustic manipulation is a contactless, and non-invasive tool[38] that has been applied to protocell formation and patterning in microfluidic environments[39]. However, to date, the full range of opportunities for artificial cell construction and control provided by the combination of these techniques has yet to be fully realised. Current technological bottlenecks in complex emulsion processing have limited the architectural complexity and functionality development of multi-compartment artificial cell models.

Here, we report the development of a droplet laboratory platform for the creation, manipulation, control, and measurement of artificial cells with distributed cores (ACDC droplet), using precision droplet microfluidics and acoustic levitation (Fig. 1a-1, a-2, Fig. S1). Membrane proteins can be incorporated in the levitated ACDC droplets, and their functions are remotely controlled in situ by the acoustic levitator. The application of programmable microfluidics and acoustics, with the incorporation of responsive bio-elements, represents a possible route to affording increased control over the compartmental organisation, connectivity, and communication, alongside opportunities for spatial and temporal control of protein function. These facets enable more routine production of artificial cells and materials with the ability to engineer more complex and life-like characteristics, more akin to their biological counterparts.

## Results

**Acoustic levitation and manipulation of ACDC droplets in air.** The ACDC droplets are defined as complex emulsion droplet materials with organised chemical encapsulations and hierarchical structures. Such a model aims to mimic eukaryotic cell compartmentalisation and its endowed functionalities. In our previous work, precursor ACDC droplets were developed as a free-standing artificial cell model that can harness membrane mimetic chemistry via encapsulated droplet interface bilayers[40]. 3D microfluidics enables the construction of polarised and directional ACDC droplets with patterned, multi-layered shells encapsulating functional nanomaterials[41]. In this work, we build upon these foundations to focus on subsequent processing methods for ACDC droplets, specifically on the internal organisation and functional control of compartment networks in three dimensions. Through this context, ACDC droplets take the form of a multisome[34] suspended in air. They are constructed using a continuous microfluidic methodology, from lipid-coated water droplets, here termed "cores", where protein-mediated communication can take place between bilayer segregated cores of the network. The production sequence of aqueous cores, together with fluidic properties, determines the network connectivity upon encapsulation. Once an ACDC droplet is dispensed from the microfluidic manifold, it can be trapped within the nodes (zones) of a vertically orientated acoustic standing wave, generated by a multi-emitter, single-axis acoustic levitator operating at 40 kHz (Fig. 1b). The droplet trapping trajectories depend on the droplet release positions (Fig. 1c and Fig. S2), and the levitated droplets typically adopt an ellipsoid shape that can be controlled by the power of the acoustic transducer array (Fig. S3). The acoustic streaming effect[42] generates convective flows within the levitated droplet, as shown by the multiphysics finite element modelling result (Fig. 1d), and these fluid dynamics may facilitate the enhancement of mass transport in each phase of the emulsion droplet along the fluid streamlines. We observed lipid bilayer formation[7,34] within the levitated ACDC droplet, where lipid monolayer-coated water cores contact each other (Fig. S4, Fig. 1e). As high acoustic impedance exists at the air-liquid interface, the amount of acoustic radiation energy penetrating into the droplet is limited, and lipid bilayers remain robust and do not fail or appear to become porous. As the total volume of the internal core network increases, the assembled droplet interface bilayer networks rotate around the vertical axis (Supplementary Movie 1) due to the induced acoustic streaming. This induced motion can be affected by external perturbation; for example, by the application of external air currents, or other asymmetric forces exerted on the ACDC droplet (e.g. applied magnetic fields). Without disruption of the acoustic standing wave field, post processing of levitated ACDC droplets in situ is possible, such as by the addition or removal of cores whilst the ACDC remains levitated in the trapping zone. Multiple ACDC droplets can be levitated and aligned at different nodes of a single standing wave field for parallel operations (Fig. 1f). By using acoustic levitation for the control of artificial cells, this system provides the ability to reproducibly process, activate, manipulate, and observe multi-compartment artificial cells with contactless methods, immediately following their microfluidic production. We demonstrate control of droplet patterning within the ACDC droplet, that lipid bilayers can be broken and reformed on demand, and that internal core and bilayer networks can be dynamically reorganised in situ. We demonstrate the ability to remotely activate/

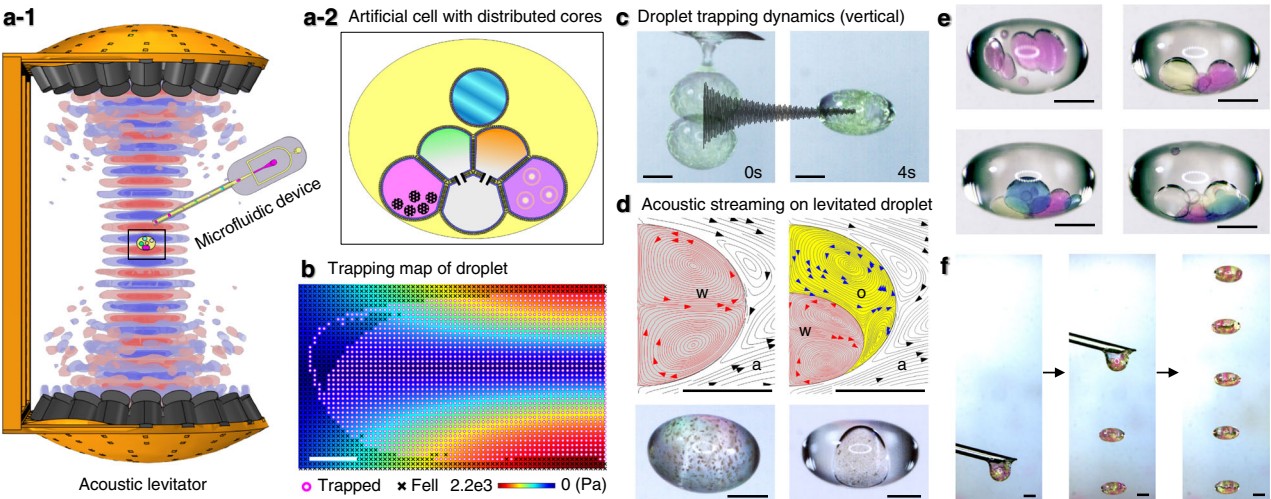

**Fig. 1 Droplet laboratory platform. a-1** Schematic of the acoustic levitator platform setup. Multiple acoustic standing waves (red and blue), generated by the acoustic levitator, can trap microfluidically formed artificial cells with distributed cores (ACDC droplet), at nodal positions. Photographs of full experimental apparatus are available in Fig. S1. **a-2** Schematic of patterned ACDC droplets encapsulating multiple reagents within the core networks. **b** Simulation results of 2 mm diameter droplet trapping in air. The pink circle (droplet levitated) and the black cross (droplet fell) symbols represent the outcome of simulated droplet release positions in the acoustic field (rainbow colour map indicates standing wave pressure profile (symmetric half)). **c** Droplet trapping dynamics: An ACDC droplet dispensed from the microfluidic device into a standing wave field oscillates vertically in the standing wave node before dampening and assuming a stable position over a period of 4 s. Additional data on droplet release positions are available in Fig. S2. **d** Top; Simulation and experimental results of the convective flows within levitated water (w) droplet in air (a) (left) and water in oil (o) double emulsion droplets in air (a) (right). Bottom; Water phases containing hydrophilic magnetic microparticles (dia. 1 μm) and minor aggregates (brown), remain suspended and circulate with the fluidic convection, without significant clustering. **e** Manually adding multiple water droplets to a levitated oil droplet with a micropipette. From top left to bottom right, pink, green, blue and clear droplets. **f** Deposition of multiple microfluidically formed ACDC droplets at sequential nodes within the acoustic standing wave field. Scale bars = 1 mm.

deactivate the mechanosensitive channel of large conductance (MscL) on demand, potentially enabling the switching on and off, of artificial cell activity. Collectively this could enable the harnessing of more complex functionality in artificial cells, providing additional control of communication networks in space and time and moving away from single-time activation.

Fused filament 3D-printing technology was employed to fabricate microfluidic manifolds with customised fluidic circuits for ACDC droplet generation. A multi-layered, 3D-printed droplet forming junction was devised (Fig. S4), the design of which compensates for the resolution limitations of fused filament 3D printing, to form uniform, size-controlled (minimum 20 μm diameter, Fig. S4), lipid-coated aqueous droplets (cores) in a continuous oil phase flow. An array of these junctions is configured within a fluidic circuit to generate multiple types of cores in parallel (Fig. 2a). Meanwhile, programmed inflow pressure profiles are applied to regulate the droplet formation on-demand, and produce droplet core sequences which template the order of reagent encapsulation (Fig. 2b). Such 1D linear core sequences lead to the packing and self-organisation of patterned 3D core networks during the ACDC droplet formation and levitation (Fig. 1c and Supplementary Movie 1). The final configuration and connectivity of core networks is collectively influenced by the microfluidic inlet flow profiles, microfluidic circuit configuration, and droplet trapping in the acoustic field. In this way, a target core containing specified reagents can be allocated to different regions of the core network for definable compartmentalisation within the ACDC droplet (Fig. 2c). This can be used for structuring the intracellular chemical organisation that is segregated by lipid bilayers with approximate control of the core's coordinates in three dimensions. Such effort may be used to control compartment connectivity and separation and thus program sequential reactions within multicompartment artificial cells.

**Contactless operation of levitated ACDC droplets**. Microfluidically formed, lipid membrane segregated, complex emulsion droplets provide metastable configurations for chemical organisation and connectivity. Meanwhile, in comparison to the batch processing methods used for single compartment protocells in aqueous environments, we present a comprehensive approach to operate individual levitated ACDC droplets and their sub-compartments, using a range of contactless methods for physical-chemical processing (Fig. 3a, Supplementary Movie 2). This may enhance the breadth of multicompartment artificial cell functionalities, enabling chemical modulations of artificial cells[43].

The first demonstration utilises a pneumatic device such that the local air flow controls the rotation of ACDC droplets (Fig. 3b). As shown in the figure, an air nozzle is carefully controlled to tangentially apply a thin air stream to the equator of the levitated droplet (Fig. 3b-i). This causes the droplet to spin about its central vertical axis whilst remaining trapped in a pressure node of the levitator. With increasing airflow, the morphology of the ACDC droplet can be changed from ellipsoid to a dumbbell shape, and eventually breaking it up into two daughter droplets, splitting as a result of droplet thinning at the central waist (Fig. 3b-ii). The shape and angular velocity of spinning ACDC droplets can be precisely controlled by the air inflow rate and stably maintained up to 35 revolutions per second as shown in Fig. 3b-iii (n = 30 for each data point, average standard error of mean = 0.53%). Whilst spinning, the surface area of interior droplet interface bilayers is minimised, and the droplet contact angle is increased until the membrane leaflets separate and the droplets fully detach. This is a consequence of the centrifugal force resulting in the dense water core moving radially outwards in the less dense oil phase (Fig. S5). For a given core network, this density-governed detachment depends upon the spinning angular velocity and droplet size, as well as the oil phase composition and the acoustic transducers power (Fig. S5). On cessation of air flow, the spinning of the

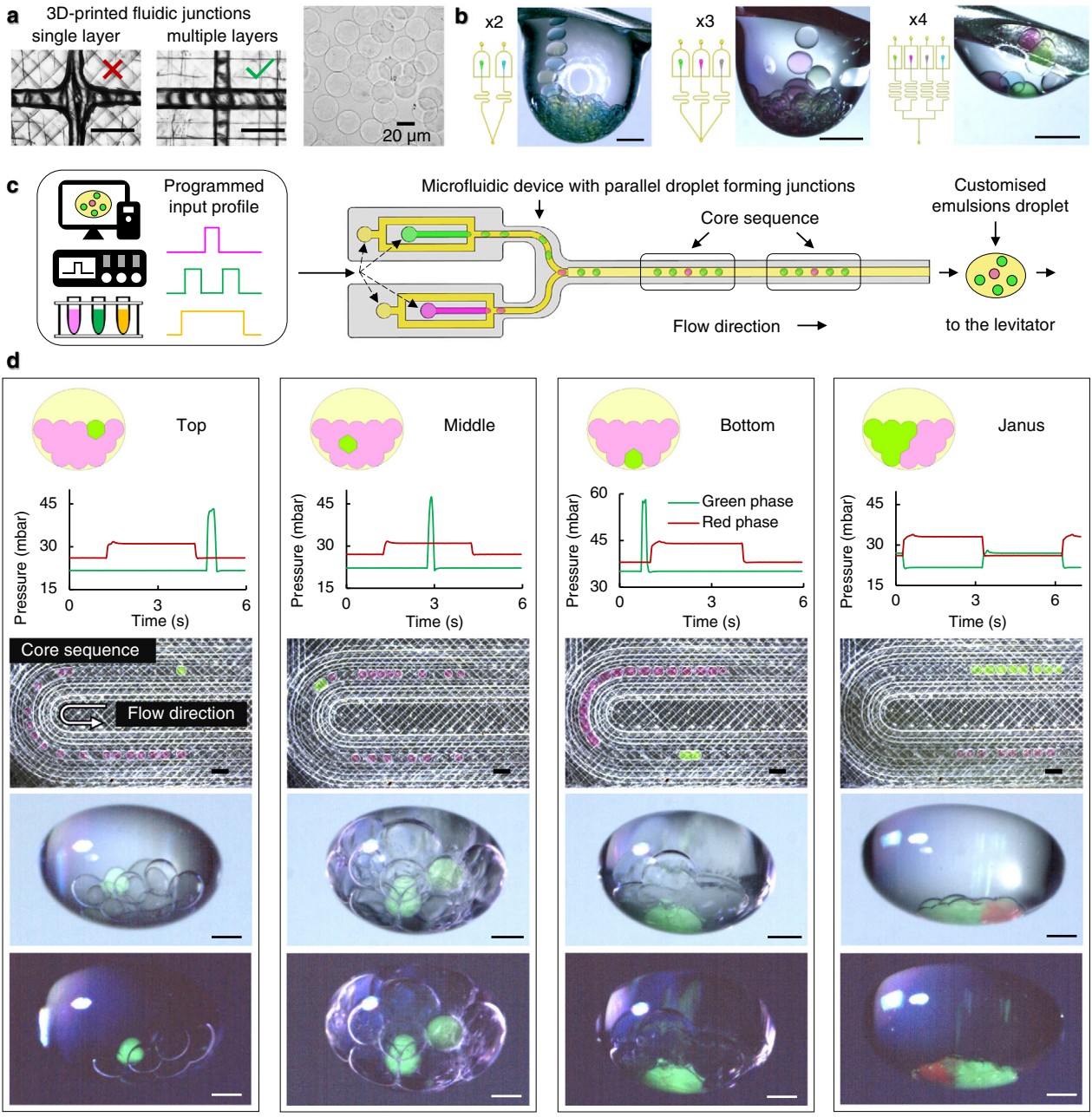

**Fig. 2 Microfluidic control of artificial cells with distributed cores (ACDC droplet) structural organisation. a** Multi-layered 3D printing minimises capillary junction size, improving droplet generation fidelity, in comparison to single-layered devices (left), see Fig. S4 for further details. Uniform, cell-sized, alginate microgels can be formed using 120 μm wide, 3D-printed, multi-layered junctions (right) (flow rates: dispersed phase: 1 ml h⁻¹ and continuous phase: 20 ml h⁻¹. Additional fluidic designs and droplet size data available in Fig. S4). **b** The addition and combination of parallel fluidic junctions enables formation of multiple different types of water droplets within an oil compartment. Increasing complexity is illustrated left to right with cores of two, three and four different chemical identities. **c** Programming of fluid inlet flows can be used to generate droplet sequences of defined order and spacing for the creation of customised emulsion droplets. Programmed input profiles regulate droplet formation at each droplet forming junction, on-demand, determining core sequences and therefore providing the chemical encapsulation template for the ACDC construct. Droplet order determines final 3D spatial arrangement of the encapsulated cores. **d** Examples of patterned, levitated ACDC constructs. The fluidic input flow profile determines the core sequence and packing order in the ACDC construct. From left to right, a specific core (green) is directed to the top, middle and bottom level of ACDC constructs, and the formation of Janus networks comprised of red and green cores. Lipid bilayers segregate cores of the internal droplet network. All scale bars = 500 μm except where otherwise stated.

ACDC droplet slows down, and the separated cores reassemble, again forming lipid bilayers between contacting cores. New core networks may be assembled in this way, with the volumes and densities of the cores, together with air flow rate, determining the release order and reassembly. This bilayer disassembly-assembly process can therefore be used to reconfigure the core network patterns within the ACDC droplet, changing the packing order. For instance, Fig. 3b-iv shows reconfiguration by this method. By controlling the fluidic input profile and sucrose concentration in two aqueous input flows, an internal network of cores that differed in size and density, (green and red droplets) were created. This network could be reconfigured with different patterns of core-core

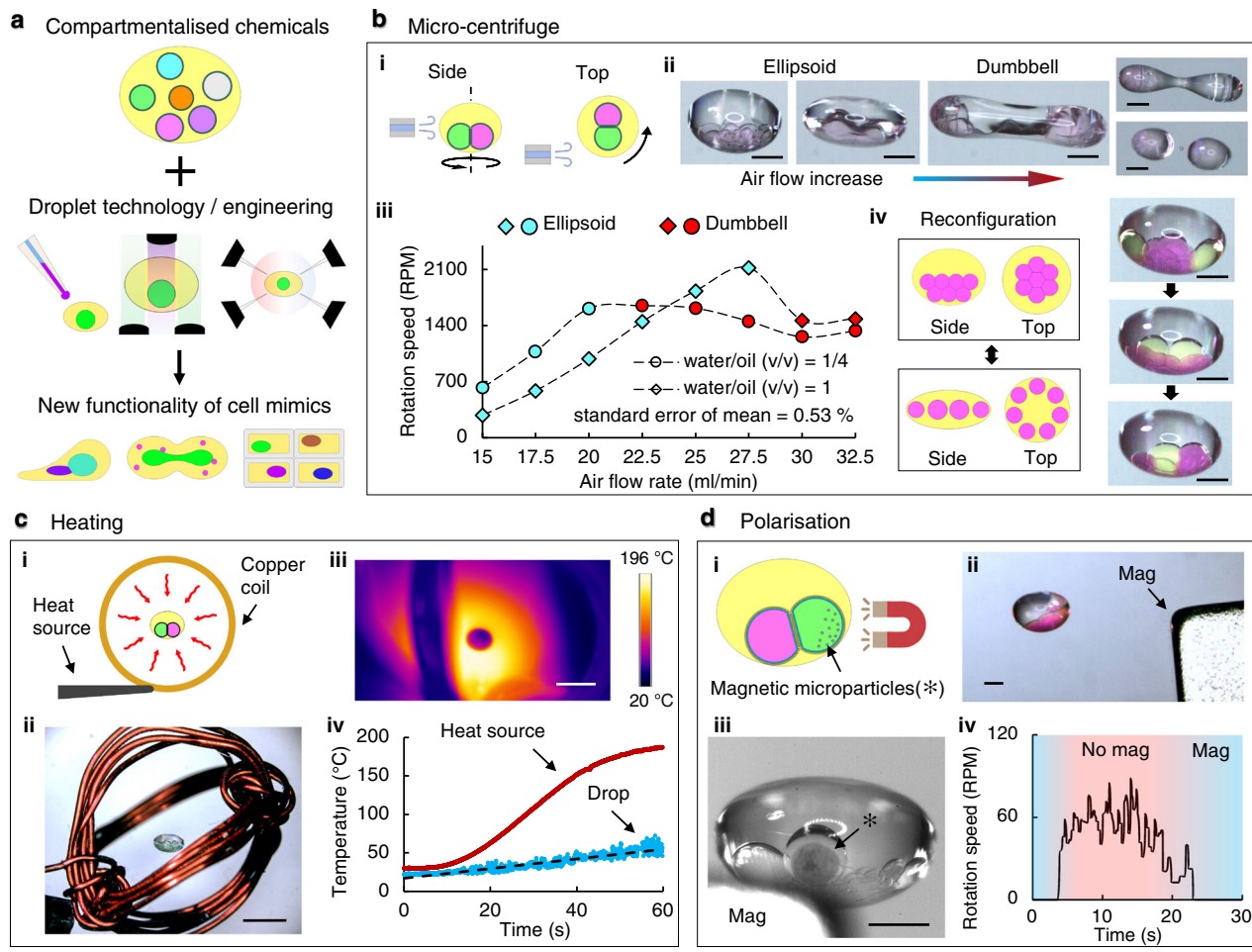

**Fig. 3 Multiple operations on levitated ACDC droplet. a** The combination of controlled chemical compartmentalisation, with subsequent droplet processing and manipulation, as methods to energise the droplet structures, can be used to impart function on the resulting materials. **b** Pneumatic operations for the spinning of levitated artificial cells with distributed cores (ACDC droplet). (i) A radially tangential air flow to the ACDC droplet rotates the levitated construct. (ii) As the air flow is increased, the droplet rotational speed increases, and the droplet becomes elongated before splitting into two daughter droplets, each with interior core network. (iii) The spinning rate can be precisely controlled by the air flow rate (inset graph: $n = 30$ for each data point, measured from a 20 s spinning interval). (iv-left column) At high rotational speeds the droplet network disassembles and lipid bilayers are separated between adjoining cores. Spin-stop cycles can be used to reconfigure the internal core network, detaching and re-assembled bilayer connections. (iv-right column) This is depicted for disassembly and reassembly of a core network (green and pink cores) being reconfigured to separate and connect different cores. Scale bars = 1 mm. **c** Thermal operations for the heating of levitated ACDC droplets. A copper wire coil was used to induce thermal energy within the levitated droplet through thermal convection (i and ii). The droplet can be heated up from room temperature to 50˚C in 1 min with gradual heating (iii and iv). Scale bars = 5 mm. **d** Magnetic operations for the manipulation of levitated ACDC droplets (i and ii). A levitated ACDC droplet containing a core with encapsulated magnetic particles (iii), can be manipulated by external magnets, providing droplet positional and orientational control (iv). The star symbol (*) indicates the position of the magnetic particles, and their gathering at the core perimeter on application of an external magnetic field. Scale bars = 1 mm.

connectivity following spin-based separation and reassembly. No core coalescence or dye mixing was observed indicating that the bilayer membranes were intact during the spinning, separation and reformation processes. Since membrane protein channels can be inserted into the bilayers of core networks, this rearrangement of connectivity can determine chemical communication and consequently function. In principle, this represents a method to reconfigure an artificial cell on demand, for the activation, or selection, of different functionalities dependent upon the precision organisation of the cores, e.g. number, size, and fluid property.

A levitated ACDC droplet can also be energised and operated remotely, utilising convective heating, photo-irradiation, or magnetic manipulation. Figure 3c-i shows the heating of droplets via thermal irradiation. Copper coils (Fig. 3c-ii) were crafted to transfer thermal energy to the levitated droplet by convective heating without disruption of the acoustic standing wave. In this way the droplet can be heated from ambient temperature to

~50 °C in one minute (Fig. 3c-iii, iv). While the temperature gradually increased, the levitated droplet started to agitate and adopt a discoid architecture, possibly due to changes in the air, oil, and water properties (e.g. density, viscosity, surface tension) in response to the heating. The disk shape of the heated droplets could be tuned back to ellipsoid by gradually reducing the input power to the ultrasonic transducers, and vice versa. This heating operation could be applied to initiate thermo-responsive reactions within the ACDC droplet. Figure 3d-i shows the employment of magnetic manipulation and the ability to control the orientation of a levitated ACDC droplet. The magnetic manipulation of droplet networks in an oil environment, through the inclusion of magnetic particles within selected droplets and the application of external magnetic fields, has been reported in previous work[44]. Here, this principle is applied whereby one specific compartment (core) of the droplet network contains magnetic microparticles and is attracted to an externally applied magnetic field whilst the

ACDC droplet is levitated. This orientates the ACDC droplet in the levitator trap (Fig. 3d-ii, iii). Consequently, the whole core network is in turn re-oriented via the adhesive forces of the membrane connections. Under the applied magnetic field, the ACDC droplet becomes slightly offset from the original equilibrium position, given the balance of both magnetic field and acoustic field forces. In this way, the levitated ACDC droplet can be prevented from passive rotation or spinning induced by other operations or disturbances (Fig. 3d-iv). The orientation of the core network can be precisely orientated (e.g. for observation) by the positioning of the external magnet in the system. In a further example of contactless processing, light-initiated reactions, such as the free-radical polymerisation of a microfluidically formed poly(ethylene glycol) diacrylate (PEGDA) droplet containing an internal oil core, is implemented in a levitated droplet (Fig. S6). The heat generated by the exothermic reaction and its subsequent dissipation can be measured in situ. This indicates the possibility to fabricate protective solid shells around soft interiors, such that the processed ACDC droplets can be stored or transported to other environments.

**Reconstitution of protein channels in levitated ACDC droplets.** Establishing controllable chemically mediated communication between compartments, provides a means to govern biochemical processes within artificial cells. In a lipid-based model, intracellular communication can be controlled by the incorporation of membrane protein channels and pores to selectively permeabilise segregating membranes. Herein, firstly we demonstrate that the lipid bilayers of the ACDC droplet remain stable and impermeable during acoustic levitation, thus preventing inter-compartmental mixing. Secondly, we demonstrate that the protein pore, alpha-hemolysin (aHL) and bacterial mechanosensitive ion channel of large-conductance (MscL), may be reconstituted into the membranes of levitated ACDC droplets to enable ionic mediated communication between selected compartments (Fig. 4). The membrane protein reconstitution was conducted either by manually adding the protein-containing cores in situ or through on-demand droplet formation of a protein-containing aqueous phase in microfluidic devices as shown in Fig. S8. The calcium-sensitive fluorescent dye, Fluo-8, was incorporated within a droplet also containing monomeric alpha-hemolysin, which is connected to $Ca^{2+}$ containing droplets within the internal core network (Fig. 4a-1). An additional Fluo-8 containing droplet without alpha-hemolysin was also connected to calcium-containing droplets and encapsulated within the ACDC droplet, serving as a non-protein-containing control. Fluo-8 fluorescence in the respective cores of the levitated ACDC droplet showed calcium flux through the membrane-spanning aHL pores, whereas the non-aHL-containing membrane remains impermeable to $Ca^{2+}$ ions (Fig. 4a-2). The observed sigmoidal fluorescent response is a consequence of an acceleration in ion-flux as an increasing number of functional heptameric aHL pores assemble on the membrane from their constituent monomers[45]. The fluorescence response then plateaus (~20 min) as the Fluo-8 becomes saturated (more discussion available in Figs. S7, S8).

Unlike the permanently open alpha-hemolysin pore that allows the diffusion of $Ca^{2+}$ following pore formation, the mechanosensitive ion channel, MscL, is typically closed but can be gated by the application of mechanical stress applied to the membrane[46,47]. A glycine-serine substitution (G22S) in the hydrophobic channel gate generates a mutant that is not spontaneously active, but possess a lower activation threshold than wild-type MscL[48,49] Interestingly, on incorporation into ACDC droplets with membranes formed of 1,2-Diphytanoyl-sn-glycero-3-phosphocholine (DPhPC) lipid (as used for aHL), no

ion-flux is observed, indicating that the MscL channels are not spontaneously activated by the acoustic levitation or any associated acoustic perturbation of the membrane network (Fig. 4b-1). The formation of asymmetric 1,2-dioleoyl-sn-glycero-3-phosphocholine (DOPC)-DPhPC membranes between contacting cores of the ACDC droplet could be harnessed to induce membrane tension and consequently spontaneous MscL channel activation (Fig. 4b-2). In these asymmetric membranes, rapid ion flux was observed via Fluo-8 fluorescence, reaching maximal response within 4 min (Fig. 4b-3). No measurable ion-flux was detected in the DPhPC symmetric membrane system with MscL (Fig. 4b-3). Similarly, DPhPC-DOPC asymmetric membranes did not give rise to ion-flux in the absence of MscL (Fig. S11). Asymmetric DPhPC-DOPC lipid bilayers were constructed by the in situ addition of a DPhPC monolayer-coated, MscL-containing core, into the levitated ACDC droplet with a DOPC-based core network. The DPhPC-coated MscL-containing core was initially prepared by incubating an MscL-containing aqueous droplet in a DPhPC-containing oil phase, to form a DPhPC monolayer-coated core, before transplanting into the DOPC-containing core network of the levitating ACDC droplet. Contact between the DOPC core network and the manually added DPhPC core, allowed for the formation of an asymmetric bilayer with one leaflet mainly consisting of DPhPC and the other mainly consisting of DOPC[50]. Once the DPhPC-DOPC bilayer is formed, the MscL channels are spontaneously reconstituted and passively activated by the tension of the asymmetric membrane. MscL activation was also achieved in the levitated ACDC droplet by the inclusion of LysoPC within the MscL-containing core of a symmetric DPhPC bilayer membrane, with LysoPC being incorporated into the inner leaflet of the lipid bilayer to induce membrane tension and protein activation[47].

**Remote control of protein function in levitated ACDC droplets to activate intracellular communication.** The ability to modulate protein channel function, on demand, represents an attractive proposition for the attainment of greater functional control over artificial cells. In this context, we demonstrate that the gating of MscL channels can be selectively activated on demand via remote perturbation using a combination of applied acoustic and magnetic forces (Fig. 5, Supplementary Movie 3). In symmetric DPhPC membranes within an ACDC droplet, the MscL channel is closed (Fig. 4b-1). The inclusion of an additional dedicated core within the droplet network containing magnetic particles, enables the orientation and 'locking' in the position of the ACDC droplet on the application of an external magnetic field (Fig. 3c).

With the incorporation of the magnetic particles in the core, in the absence of a magnetic field (non-locked state), the MscL channels remain closed (0–30 s Fig. 5b, c). The MscL channels are only activated upon application of an external magnetic field to 'lock' the droplet in position and continuous $Ca^{2+}$ ion-flux transport is observed (30–240 s Fig. 5b, c). The fluorescent response rapidly proceeds to a maximum within ~3 min of activation, indicating a comparable extent of channel opening to that observed with the formation of a DPhPC-DOPC asymmetric membrane. In the absence of MscL, no fluorescence intensity increase is observed in the Fluo-8 containing core of a locked construct (Fig. 5d), evidencing the membranes remaining intact and the ion-flux being protein mediated (Fig. S9). By the selective application of the external magnetic field, to move between 'locked' (MscL active) and unlocked/rotating (MscL inactive) states, it is possible to repeatedly control MscL opening and closing in the ACDC artificial cell (Fig. 6a). Unlike chemical activation, this approach enables deactivation and repeat activation, creating opportunities for multiple use switches, analogous

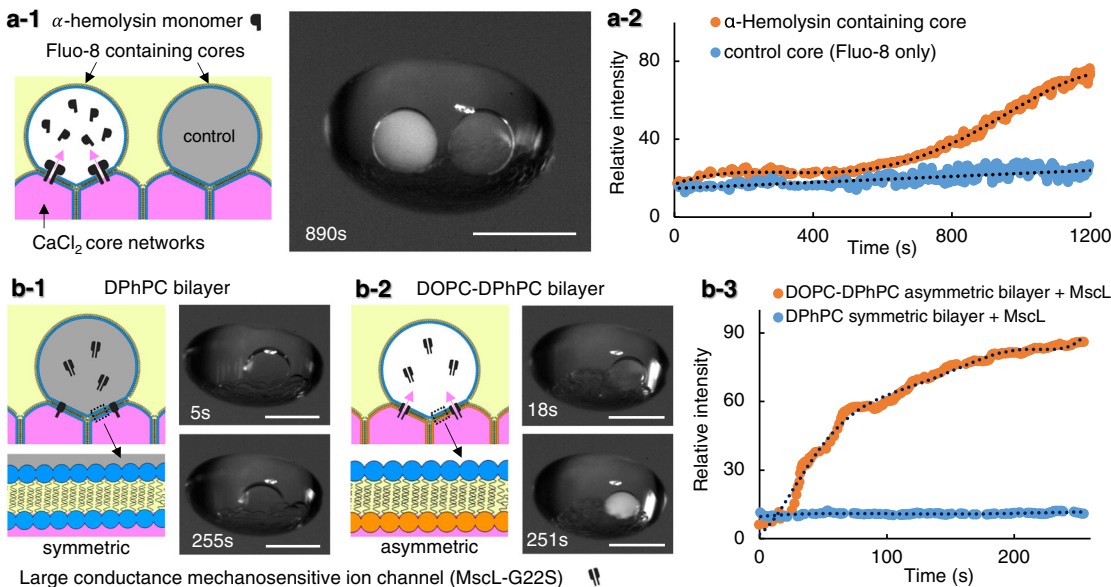

**Fig. 4 Reconstitution of functional alpha-hemolysin pores and bacterial MscL protein channels into the artificial membranes of the core network of levitated ACDC droplets. a-1** Alpha-hemolysin (aHL) monomers contained in an internal core bind to the membrane separating neighbouring droplets and assemble into heptameric protein pores in the DPhPC bilayer. $Ca^{2+}$ ions diffuse through aHL pores moving from the source cores to the protein-containing core where fluorescence is induced by the presence of the calcium-sensitive fluorophore, Fluo-8. No ion-flux is observed in an identical control droplet without aHL monomer. **a-2** The graph indicates the fluorescence intensity over time of both aHL containing and non-aHL containing (control) droplets in the levitated artificial cells with distributed cores (ACDC droplet) over 20 min. (Dotted lines = moving average). **b** The mechanosensitive channel, MscL, opens in response to membrane tension. Gating of MscL channels in levitated ACDC droplets with membranes of different bilayer leaflet compositions. **b-1** In symmetric DPhPC bilayer networks no $Ca^{2+}$ flux is observed, the channel remains closed; **b-2** In asymmetric DOPC-DPhPC bilayer networks asymmetric membrane tension is induced, opening MscL channels, and giving rise to $Ca^{2+}$ flux and observed fluorescence. (No fluorescence increase is observed in the absence of MscL (SI)) **b-3**, Time-course fluorescence in symmetric (DPhPC) and asymmetric (DOPC-DPhPC) membranes with incorporated MscL as depicted in **b-1** and **b-2**. Asymmetric membrane tension-induced activation of MscL channels enables $Ca^{2+}$ flux, eliciting maximal fluorescence response over the course of 250 s (orange trace). No fluorescence increase is observed in the symmetric membrane system (blue trace) indicative of channel inactivation, as well as the asymmetric membrane without MscL reconstitution (Fig. S11). All scale bars = 1 mm.

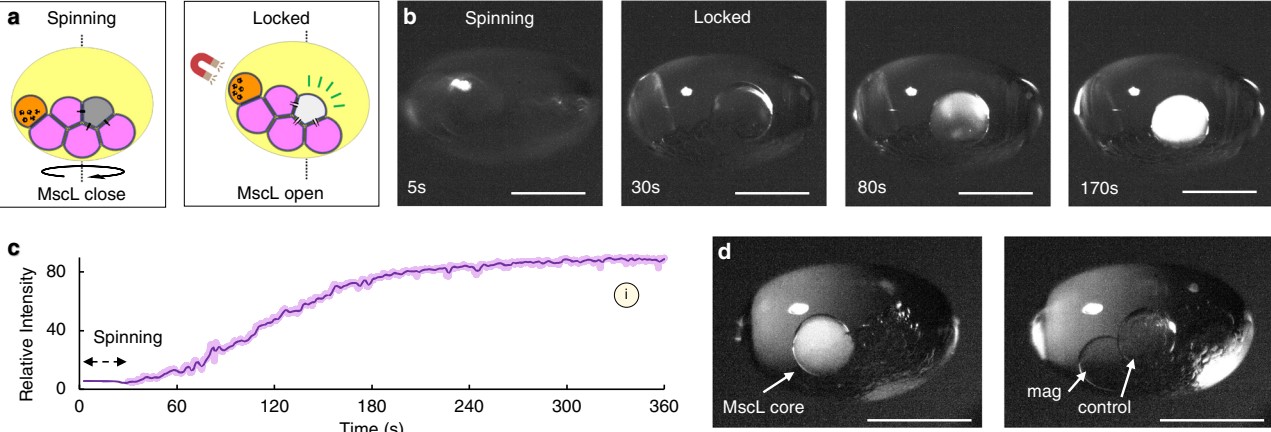

**Fig. 5 Remote control of ion channel gating in levitated ACDC droplets core networks using magnetic manipulation. a** Experimental concept: The incorporation of a core containing magnetic particles enables switching of the levitated artificial cells with distributed cores (ACDC droplet) from a freely rotating to a locked state on the application of an external magnetic field. In the free (spinning) state acoustic forces are dissipated by subtle movement. In the locked state, this is dampened and instead manifests as induction of membrane tension. This may be used to selectively activate MscL channel gating. **b** Time sequence images of MscL gating in a levitated ACDC droplet. From left to right; $t = 5$ s in the absence of an applied magnetic field the ACDC droplet is allowed to spin in the acoustic trap. At $t = 30$ s, the magnetic field is applied, locking the ACDC droplet in position. MscL is activated and $Ca^{2+}$ ion flux induced, observed by the fluorescent reporter, fluo-8, present in the MscL-containing core. At $t = 80$ s, significant fluorescence increase is observed. With the fluorescence response saturated at $t = 170$ s. In the absence of MscL no fluorescence increase is observed (Fig. S13, DPhPC symmetric bilayers). **c** Fluorescence intensity trace of $Ca^{2+}$ flux through MscL channels into a fluo-8 containing core of a locked, levitated, ACDC droplet. (Same experiment labelled (i) in Fig. 6b). (solid line = moving average). **d** Images of a levitated ACDC droplet with a MscL-containing core (left), a magnetic particle-containing core and a control core containing Fluo-8 only (no MscL) (right). The orientation of the droplet here is controlled by the placement of an external magnet to enable image acquisition from two angles to illustrate the three highlighted cores. Scale bars = 1 mm.

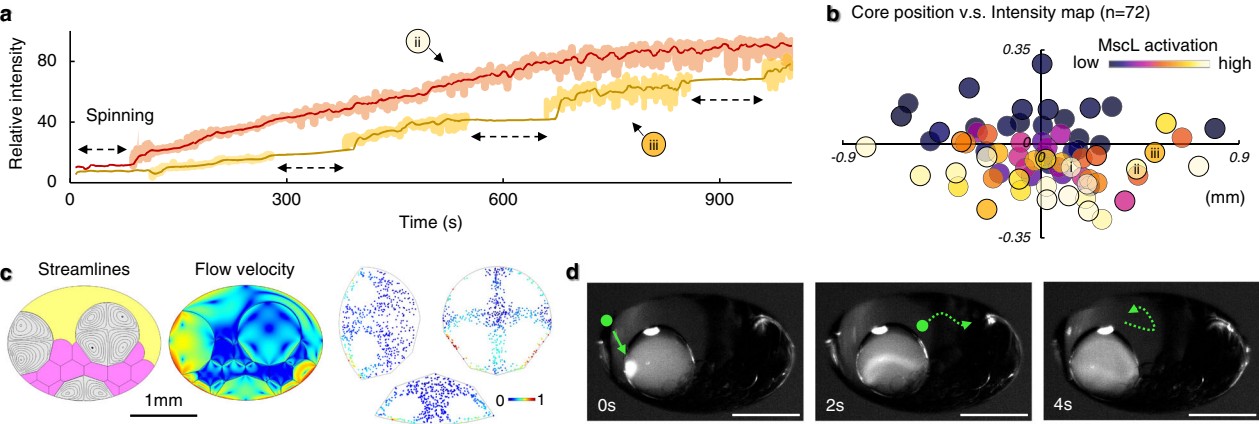

**Fig. 6 MscL gating behaviours in levitated ACDC droplets. a** Repeat activation and deactivation of MscL gating in levitated artificial cells with distributed cores (ACDC droplet) can be achieved by repeated lock/spin operations. The orange trace shows an ACDC droplet experiencing one cycle of spin/lock operation, enabling ionic communication (Experiment labelled (ii) in **b**). The yellow trace shows an ACDC droplet experiencing four sequential cycles of spin/lock operation, enabling repeated activation and shut-down of ionic communication within the ACDC droplet. The relative fluorescence intensity did not increase during the spin intervals (each ~100 s), indicating the MscL channels were mostly deactivated during these periods and activated on application of the magnetic lock. (Experiment labelled (iii) in **b**). **b** Relationship of core position and MscL activity: Heatmap depicting the relative rate of ion flux (rate of fluorescence increase) for MscL-containing cores and their positions within different ACDC droplets (x/y offset from ACDC droplet centre). The plot origin is the centroid of each levitated ACDC droplet, and each circular data point represents a separate experiment with the circle's coordinates corresponding to each MscL-containing core position relative to the centroid of the host ACDC droplet. Marker colour indicates relative activity. Markers i, ii and iii correspond to the individual traces shown in Figs. 5c and b. **c** Finite element modelling of acoustic streaming induced convective flow patterns in the cores (grey) at different positions within a locked, levitated ACDC droplet. Left: convective flow streamlines in the cores. Middle: flow velocity magnitude (logarithmic scale) within the ACDC droplet. Right: Particles (5 μm) distribution under the convective flow within the cores. These simulations show how acoustic streaming profile varies with core position, corroborating experimental findings in **b**. **d** Experimental time sequence images over 4 s show local Ca$^{2+}$ release through MscL channels at the membrane followed by dissipation of the fluorescence with acoustic streaming flow (corresponding to flow simulations in **c**). Such hot-spots of MscL activation were observed stochastically in regions and periods of high induced local tension. Green spots and arrow dotted lines indicate the flow direction. Scale bars = 1 mm.

to that of biological cells. This is in contrast to the one-time only activation typically employed in artificial cell communication, and processes initiated on the addition or liberation of activating species. This selective on/off activation is possible due to the inability of the magnetically 'locked' droplet system to effectively redistribute the acoustic irradiation stress. This force would otherwise be minimised through small changes in droplet orientation and position. Acoustic irradiation stress has previously been applied to deform lipid vesicles and small living organisms to measure the membrane Young's modulus[51]. Boundary-induced acoustic streaming effects have also been studied, indicating that steady-state torques can be induced on levitated samples by applying asymmetry in the system[42]. In our experiments, the asymmetry from the droplet network itself, as well as other environmental perturbations or operational forces, exert a torque on the ACDC droplet. Without magnetic 'locking', such torque induces the subtle movements of the levitated ACDC droplet, observed as re-centring (Fig. 1b, c), convective flows (Fig. 1d), and slow rotation (Fig. 3c), due to the low air friction within the system. In the 'locked' configuration, such energy cannot readily be converted to the kinetic motion of droplet rotation, but instead, generates membrane tension within the lipid bilayers of the core network, this tension force being sufficient to activate the incorporated MscL channels (Fig. 5).

MscL channels are known to respond to membrane tension in the region of 10–12 mN m$^{-1}$ [52–54]. Multiphysics simulation of fluid dynamics resulting from acoustic streaming enables the quantification of shear forces exerted on membranes of the droplet network in a magnetically locked, levitated ACDC droplet. These simulations indicate that in the magnetically locked state the acoustic standing wave induces tensions in the region of 8–16 mN m$^{-1}$ on the membranes of the levitated droplet network (Fig. S12), which is notably in the region of the membrane tension

required for MscL activation. Under the magnetic operating conditions, the greatest experimentally measured MscL activity corresponded to ~88% of that observed in DOPC-DPhPC asymmetric bilayer conditions (Fig. 4b), indicating the establishment of similar membrane tension levels. The combined magnetic lock and acoustic activation of the MscL channel were also found through both modelling and experiment, to be influenced by the position of the protein-containing core within the levitated ACDC droplet (Fig. 6b). The simulated fluid-dynamic profile of acoustic streaming within different cores of an ACDC droplet, indicates that fluid shear is greatest in the membranes of the droplets below the horizontal mid-plane and when closer to the droplet perimeter (Fig. 6c, Supplementary Movie 3). With this model of shear-induced membrane tension, the extent of MscL activity may be predicted by the local fluidic shear tension induced by the acoustic streaming effect, across different locations of the ACDC droplet. This model is corroborated experimentally where MscL activity was systematically measured with the protein-containing core in different locations of the ACDC internal droplet network (Fig. 6b). MscL does not tend to activate when the protein-containing core is close to the top or within the mid half of the ACDC droplet, whereas greater MscL activation is achieved as the protein-containing core moves through the bottom half or when near the perimeter of the ACDC droplet. The manifestation of acoustic pressure through the ACDC droplet depends upon the applied acoustic field, the arrangement of aqueous and oil phases, and the membrane barrier. In these demonstrations a relatively simple uniform acoustic standing wave is used. However, by shaping the acoustic field with consideration of the ACDC droplet structure (e.g. by independently addressing solenoids of the multi-emitter source) it should be possible to target activation to other regions.

In addition, it was further observed that rapid high intensity Ca$^{2+}$ fluxes originated at local positions on the membranes as an

indication of mass MscL gating (Fig. 6d). The resulting $Ca^{2+}$ induced fluorescence then dissipated through the protein-containing core, following the convective flow induced by the acoustic streaming effect (Fig. 6d). These flows are compatible with simulated convection and tracer microparticle distributions (Fig. 6c). We again rationalise that the stochastic nature of these mass-activation events is, in part, due to the interplay of precise membrane position, local variations in acoustic pressure and the ability of the wider core network to redistribute induced tension, creating transient local 'hot-spots' of high channel activation. This phenomenon was not observed in the aHL group, where ion-flux is a consequence of the more homogeneous passive diffusion of ions through the formed and permanently open aHL pores, which is not modulated in this way. In principle, this MscL gating behaviour could be used to control the generation of chemical gradients, whilst the internal acoustic streaming may govern the mixing or transport dynamics of encapsulated contents in specific cores for selected chemical reactions. These may afford possible opportunities for core network function not achievable by passive diffusion and slower chemical equilibration. In the MscL activation experiments, whilst the rate of fluorescence intensity increase depends upon the membrane position and application of the dual magnetic and acoustic perturbation, it is also influenced by the ionic gradient and the exact reconstituted protein densities, which is itself a function of bilayer area, droplet volume, protein concentration and reconstitution efficiency[55], with the latter expected to contribute to experimental variability. As the research field seeks a better understanding and regulation of protein reconstitution, control over these variables, alongside core network architecture and acoustic field refinement, indicates that more sophisticated supervision of ion signalling via controlled activation/deactivation of MscL should be achievable.

Overall, these data-driven interpretations indicate that the presented MscL gating control, can likely be optimised with further refinements in predictive modelling and the control of core network organisation (e.g. core size, position and connectivity). In combination, these should be used to tune other input parameters, such as fluid properties and the acoustic radiation force field levels and shapes. This emphasises the continued importance of precision control in chemical compartmentalisation, which is underpinned by microengineering technologies, and has become an essential tool for the development of functional artificial cells.

## Discussion

In summary, we present a platform for the creation, manipulation, control and measurement of complex emulsion-based, multi-compartment, artificial cells (ACDC droplets) at single compartment (core) resolution, using droplet microfluidics and acoustic levitation. The sizes, reagent encapsulations, and 3D organisation of cores can be controlled by programmed microfluidic circuits and inlet flow profiles, with droplet sequence determining 3D droplet network connectivity. The levitated artificial cell droplets can be operated in situ to reconfigure the compartmentalised droplet network in the acoustic levitator, orientate the structure on-demand, and apply thermal and light energies. This is demonstrated by pneumatic spinning, magnetic manipulation, thermal convection and free-radical photopolymerization, respectively. We show that lipid bilayers, segregating the internal compartments of the artificial cell, are robust and leak-free in the acoustic field. Furthermore, we demonstrate the ability to reconstitute alpha-hemolysin and MscL membrane proteins, to direct $Ca^{2+}$ ion transport across membranes of the levitated compartmentalised droplet network. Importantly, the gating of incorporated MscL channels can be selectively and repeatedly controlled to activate/deactivate channel activity through the application of

an external magnetic field alongside the levitating acoustic field. In this way, membrane tension can be perturbed to induce channel opening. This establishes remotely controlled intracellular communication within the artificial cell, which can be repeatedly switched on and off. Such a combination of acoustic and magnetic field activation could be used to activate artificial cells in situ for future diagnostic and therapeutic applications or used to initiate and halt artificial cell functions. The components of the presented experimental platform are all fabricated using accessible 3D-printing and have robust performance in diverse settings, hence this approach is both generalisable and reconfigurable for different chemical and synthetic biology laboratories.

In this report, ACDC droplets were employed to study the control of chemical-mediated communication, via reconstituted membrane proteins, within the multicompartment droplet network model. Generally, the size of such models ranges from >10 μm up to a few millimetres. These lipid-segregated droplet networks can display cellular-like functions and emergent behaviours, finding utility in biotechnology applications and as useful soft matter, semi-autonomous materials, such as artificial nerves[56] and biological logic circuits[30], with practical applications not necessarily constrained to materials on the scale of biological cells[57]. Meanwhile, smaller objects can be efficiently levitated in air by higher frequency acoustic standing waves[58]. Whilst the ACDC droplets levitated here are on the order of 1–3 mm in diameter with internal cores typically in the range 20–300 μm, advances in 3D-printing will afford opportunities for miniaturisation of 3D-printed microfluidics and the ACDC droplets they produce, surpassing the minimum 20 μm diameter internal cores reported here. Even smaller objects, such as microparticles and nanomaterials, can be encapsulated in the cores and serve as functional units for high-order activities. Acoustic manipulation has been applied to objects of this size and used to control protocell and living cell interactions in microfluidic chambers that are suitable for integration with high-resolution imaging[59]. In addition, aerosol and particulate down to 1 μm can be captured and evaluated using an acoustic levitation method[60]. Meanwhile, the levitated object can be stabilised with a shaped acoustic field[61], or approaches comparable to the magnetic positional lock technique employed here for more stabilised imaging. These approaches could also be applicable for higher-resolution imaging of artificial cells like those reported here. With further instrument development, it should be possible to precisely operate smaller ACDC droplets in air; for example, with the control of free movement in three dimensions, underpinned by the further study and modelling of the physics and fluid dynamics of levitated objects[62].

As the strategy of synthetic biology shifts from exploratory research to systematic quantified study, precision engineering and data-driven methods will play a key role for the development of increasingly sophisticated artificial cells[63,64]. Structuring intracellular compartments of artificial cells with organised chemistries is an important step in the bottom-up creation of artificial cells, thereby affording increased functionality. The integration of droplet forming, sequencing, and encapsulating microfluidics alongside post processing, affords a greater control space for the fabrication and operation of individual complex emulsion droplets as artificial cells. Following in situ processing, such droplets can be subsequently 3D-printed to construct more complicated, tissue-like super-structural materials. This work provides a versatile platform technology for engineering multicompartment artificial cell chassis, that bridge biomimetic materials and artificial cell functionality development. With the developed platform, it is possible to investigate and operate artificial cells at varying length scales, ranging from the building blocks, such as proteins and lipid bilayers, to the droplet network organisation and architecture, whilst the function and morphology can be

altered, or activated, repeatedly, providing analogous processes to those of natural cells. The ability to activate and deactivate membrane proteins to govern reaction or chemical communication pathways, on demand, provides an approach to the more routine use of artificial cells as next generation smart materials. In this context, the demonstrated capability will enable the design of programmable cell behaviours, incorporating both multiphysics engineering and (bio)chemical approaches. For example, it is envisaged that molecular assembly and biochemical synthesis[65–67] could be incorporated and integrated with the polarisation, reconfiguration and remote activation demonstrated. These will enhance our capability to create and control artificial cells and synthetic biological systems[68–70], towards the development of more sophisticated, life-like materials.

## Methods

**Chemicals and components**. Alginic acid sodium salt, calcium chloride, potassium chloride, silicone oil AR20, hexadecane, glucose, HEPES, EDTA, PEGDA, sulforhodamine B, calcein, lissamine green, alpha-hemolysin (aHL) monomer and n-dodecyl β-D-maltoside (DDM) were all purchased from Sigma-Aldrich. Fluo-8H AM was purchased from ATT Bioquest. BioRad Chelex 100® resin was purchased from BioRad. Hydrophilic, magnetic silica microparticles (SiMAG/MP-DNA 1.0 μm) were purchased from Chemicell. 1,2-Dioleoyl-sn-glycero-3-phosphocholine (DOPC) and 1,2-diphytanoyl-sn-glycero-3-phosphocholine (DPhPC) lipids were purchased from Avanti Polar Lipids. Neodymium magnets were purchased from RS Components.

The preparation of the precursors for the ACDC droplets was as follows. Water phase: potassium chloride was dissolved in deionised water at 0.15 M concentration with 0.01 M HEPES and 150 μM water soluble dyes (sulforhodamine B, calcein, or lissamine green). Buffer was filtered with Nylon membrane filters (0.22 μm, Fisherbrand) for use in the experiment and stored as a stock solution. Oil phase: Lipid powders were dissolved in hexadecane at 20 mg mL⁻¹ as stock, and mixed with additional hexadecane and silicone oil at different ratios for microfluidics experiments. To prepare the microparticle containing aqueous phase solution, the original particle-containing solution (from supplier at 50 mg mL⁻¹) was mixed in the prepared 0.15 M KCl buffer at a ratio of 1:100 v/v. To prepare the alginate solutions, alginate powder was added to 0.15 M KCl buffer at 2% w/v, and mixed using a magnetic stirrer (IKA RCT basic safety control stirrer) agitated at 800 rpm, and at 50 °C, for 4 h. All prepared solutions were kept for a maximum of one week for use in experiments.

**MscL-G22S expression and purification**. The MscL-G22S mutant expression construct was made from the wild-type MscL 3.1 construct using QuikChange site-directed mutagenesis kit (Agilent Technologies, Santa Clara, CA, USA)[48]. To purify the channel protein, MscL-G22S was expressed in BL-21 (DE3) (Novagen) E. coli strain, which was grown at 37 °C to OD600 0.8. The protein expression was induced with 1 mM IPTG for <1 h as previously described[71]. Briefly, the cell pellet was suspended in PBS in the presence of ~0.02 mg mL⁻¹ DNase (Sigma DN25) and 0.02% PMSF (Amresco M145) and disrupted using a TS5/48/AE/6 A cell disrupter (Constant Systems) at 31,000 psi at 4 °C. Cell debris was removed by centrifugation at $12,000 \times g$ for 15 min at 4 °C. The membranes were pelleted at 45,000 RPM in a 45 Ti rotor (Beckman) for 3 h at 4 °C. The pellets were then solubilized in PBS with 8 mM DDM overnight at 4 °C followed next day by centrifugation at $12,000 \times g$ for 20 min at 4 °C. The supernatant containing the solubilized MscL G22S protein was incubated with cobalt sepharose (Talon®, 635502, Clontech) followed by several washes with PBS supplement containing 15 mM Imidazole (Sigma, 56750). The MscL protein was then eluted with 500 mM imidazole PBS and concentrated using a 100 kDa Amicon-15 centrifugal filter unit (Merck Millipore) by diluting imidazole with DDM PBS prior to centrifugation. Protein concentration was estimated using polyacrylamide electrophoresis with SimplyBlue™ (LC6065, Thermo Fisher) staining.

**Preparation of aqueous droplet solutions for protein-related experiment**. Calcium chloride solutions used in the protein-mediated communication experiments consisted of 0.89 M CaCl₂ and 10 mM HEPES, adjusted to pH7.4. Protein-containing and non-protein-containing control droplets were made up from a base solution consisting of KCl, sucrose, EDTA and HEPES adjusted to pH7.4. BioRad Chelex 100 resin at ~5% w/v ratio was added to the base solution to remove any divalent cations which might affect the background fluorescence values on the addition of Fluo-8H. The mixture was agitated on a tube roller for 2 h before being filtered through a 0.2 μm filter to remove the Chelex resin. To this Fluo-8H was added either alone, or along with either aHL, MscL1-G22S, or DDM (the detergent used to stabilise purified MscL) in the case of control droplets. The solution was made up to a final concentration of 1 M KCl, 0.6 M sucrose, 5 μM EDTA, 10 mM HEPES, 55.8 μM Fluo-8H and either 155 nM aHL monomer (alpha-hemolysin droplets), 4 μM DDM (DDM only control droplets) or a 1:1000 dilution of MscL1-G22S stabilised with 1 mM DDM (providing a final DDM concentration in

experimental droplets of 1 μM DDM) (mechanosensitive droplets). The KCl/sucrose concentrations used ensured that protein/control droplets were osmotically, but not ionically, matched to the CaCl₂-containing droplets. For the droplet network reconfiguration experiment, KCl buffers were manually prepared with different concentrations of sucrose to control the density of microfluidically formed droplets. The droplet size was controlled by the input flow rate to the microfluidic devices.

**Droplet laboratory setup**. Microfluidic devices were designed and modelled with COMSOL Multiphysics (the dimensions of tubular channels are in the range of 120–500 μm diameter), and were printed using fused filament fabrication printers (Ultimaker 5) with cyclic olefin copolymer filament (Creamelt), using 0.25 mm AA print-cores. The print g-codes were programmed using Cura (version 4.8.0) software with customized settings. The layer height was controlled at 0.06 mm, and the printing speed was tuned at 20 mm s⁻¹ with default printing temperatures (255 °C). Ultimaker material station and air filtering system were used to maximum the printing quality. Droplet content precursors were loaded in syringes (0.5, 1, 2.5, 5 mL, gas-tight, SGE Analytical Science), and were delivered to the microfluidic devices at a constant flowrate, through PEEK and PFA interconnects and FEP tubing, using syringe displacement pumps (KD Scientific, model 789200L). Precursors were also loaded in 15 mL Falcon tubes, and were delivered to the microfluidic devices using controllable pressure profiles, programmed by wave functions using ELVEFLOW pressure pumps (OB1 MK3+).

For the levitation of ACDC droplets, a multi-emitter, single-axis acoustic levitator was used, with an array of Murata MA40S4S transducers of 40 kHz frequency, which generates acoustic waves in air at a wavelength of 8.65 mm. The driving signal originates from a driver board circuit with an Arduino Nano (to generate square signals) and an L297N amplifier. More information on the electronics can be found in the original TinyLev article[38]. The TinyLev device is composed of two oppositely facing arrays of 36 transducers each (72 in total) and are arranged in 3 rings of 6, 12 and 18 transducers, forming a hexagonal pattern. The distance between the top and bottom arrays of transducers (trapping region) was 11 cm. The transducers are fixed on a 3D-printed skeleton of the TinyLev. Once this device is assembled, the transducers can transform the electric signal received from the driving board circuit into acoustic power, which causes trapping of the ACDC droplets.

The fluidics, acoustics, optics and other operational elements were fixed on an optical board (30 × 30 cm) with 3D-printed stands and holders, as the prototype of the droplet laboratory. The linear movement of the parts were manually controlled by micromanipulators and lab-jacks. The pneumatic device was composed of a 3D-printed stand to fix 30 cm of plastic tubing (inner diameter 0.5 mm) and the nozzle was placed 3 cm away from the levitated droplets. Air was loaded in a 60 mL plastic syringe connected to the tubing, and the air inflow rate was controlled by a syringe pump. The magnetic manipulation device was composed of a 3D-printed stand to mount a metal bar, to which a 1 cm³ cubic neodymium magnet was attached at the end. The heating element was composed of customised copper coils, made from copper wire (1 mm diameter), attached to a soldering iron of which the temperature can be controlled between 100 and 400 °C. This was mounted on a metal pole. A UV torch (365 nm wavelength) was used for the PEGDA photopolymerisation reactions.

The imaging setup for general experimental images and videos was composed of a horizontally placed Nikon SMZ745T microscope with a mounted high-speed camera (MegaSpeed, up to 1300 frame per second with 640 × 480 pixels), a USB digital camera with default tele tube placed at 45 degree, and three light sources, including one RGB LED lamp (RS), one white light LED bulb (Thorlabs), and one LED ring (default with USB camera). A thermal imaging camera (Micro-Epsilon) was used to capture the temperate profile. All the components were placed at fixed positions during the experiment. A Nikon AZ100 and a Nikon MM 800 were used to take the images of the droplets flowing in microfluidic devices and the images of formed alginate microgels in a Petri dish filled with buffer, respectively.

**Fluorescence imaging**. A custom built optical set up was added to the Nikon SMZ745T stereomicroscope in order to carry out fluorescence imaging experiments. This consisted of an imaging arm with two $F = 100$ mm lenses. The first placed ~100 mm from the c-mount of the stereomicroscope served to collimate light exiting the stereomicroscope. This provided an infinity corrected region of the light path into which a bandpass filter centred at 525 nm with a width of 50 nm (Edmund optics) was placed. The second $F = 100$ mm lens was located after the filter ~100 mm from the exterior housing of a Basler Pulse pu1280-54 um camera. Image capture was carried out using Basler's pylon viewer software V6.2 with global shuttering on, an exposure time of 10 ms and an acquisition rate of 25 Hz. Illumination was achieved by selective use of a blue LED array as part of the RGB LED light source (RS). Lenses, lens tubing and other optomechanics elements were purchased from Thorlabs.

**Data analysis**. Images and videos were analysed using Nikon Element software and the FIJI distribution[72] of ImageJ software with customised coding of Macros. Thermo profiles were analysed by TIM Connect software. Numerical data were processed with MATLAB (R2020a) and Microsoft Excel.

**Multiphysics simulation**. Computational models of the TinyLev acoustic field were achieved using COMSOL Multiphysics software (version 5.4). The structure of TinyLev (in the format of an STL file) was imported into COMSOL for the 3D simulation. The distribution of the acoustic field within the trapping region was solved using the Pressure Acoustics, Frequency Domain interface, while trapping of a 2 mm aqueous droplet in air was modelled using the Particle Tracing for Fluid Flow Interfaces. The Pressure Acoustics, Frequency Domain Interface solves Helmholtz equations for given frequencies, detailed in the COMSOL Multiphysics acoustic module user's guide, and within the frequency domain, the Helmholtz equation becomes Eq. (1):

$$-\frac{1}{\rho}\nabla^2 p_t - \frac{k_{eq}^2}{\rho}p_t = 0 \tag{1}$$

$$p_t = p + p_b$$

$$k_{eq}^2 = \left(\frac{\omega}{c}\right)^2$$

where $p_t$ is the total acoustic pressure, $k_{eq}$ is the wave number, $\rho$ is the density, and $c$ is the speed of sound. However, no background pressure field was added to the model, hence $p_t = p_0$, where $p_0$ is a pressure variable defined within the interface.

When introducing the particle tracing for fluid flow interface for droplet trapping, it allows interaction between the acoustic field distribution from the first model (pressure acoustics, frequency domain) and particles introduced from the particle tracing interface.

For the acoustic streaming simulations, a 2D geometry was prepared, including models of the ACDC droplet domain and sub-compartments. Initially, the model uses the thermoviscous acoustics, frequency domain interface, which solves for the first-order acoustic fields ($p_1, u_1, T_1$). The boundaries of the ACDC droplet defined in this model will be responsible for the streaming effects. Next, the laminar flow interface is added to solve the time-averaged net flow[5], driven by the first-order acoustic fields as shown in Eqs. (2) and (3):

$$\nabla \cdot (\rho_0 u_2) = -\nabla \cdot (\langle \rho_1 u_1 \rangle) \tag{2}$$

$$\nabla \cdot \sigma_2 = \nabla \cdot (\rho_0 \langle u_1 u_1^T \rangle) \tag{3}$$

where the terms with subscript 1 belong to the first-order acoustic field solution and terms with subscript 2 belong to the streaming flow model. The first-order terms (R.H.S) are introduced into the Laminar Flow model as weak contributions in domains and boundaries. To complete the simulation of the streaming effect within the ACDC droplet, Stokes drift contributions are included on the boundaries responsible for the streaming effect[73], in the form of Eq. (4):

$$u_2 = -\langle (s_1 \cdot \nabla) u_1 \rangle \tag{4}$$

$$s_1 = \frac{u_1}{i\omega}$$

Finally, particles of 5 μm diameter were released within certain cores using the Particle Tracing for Fluid Flow interface and their trajectories investigated under the influence of the acoustic streaming effects[74].

The tension exerted on the levitated droplet network, which is induced by the acoustic streaming effect, was calculated as the integral of the multiplication of fluidic shear rate and the fluid viscosity based on the models shown in Fig. S12.

Computational fluid dynamic modelling was carried out in COMSOL Multiphysics, using a microfluidic module as described in our previous study[75].

## Data availability

The data that support the findings of this study are available within the main text of this article and its Supplementary Information. Any other relevant data are available from the corresponding authors upon request.

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

## Acknowledgements

This work was primarily supported by funded by the European Horizon 2020 project ACDC (Artificial Cells with Distributed Cores) under project award number 824060 awarded to D.A.B. and O.K.C. at Cardiff University. The custom-built fluorescent imaging setup and development was in part supported by a Cardiff University MRC Confidence in Concept award (MC_PC_17170) awarded to O.K.C. B.W.D. would like to acknowledge financial support from the Wolfson Foundation and the Royal Society. P.R. and B.M. would like to acknowledge support from the Australian Research Council.

## Author contributions

J.L. conceived and led the research, and performed all the experiment. O.K.C., B.W.D. and D.A.B. equally contributed to the experiment plan designed by J.L. D.A.B. contributed to the provision of microfluidics. W.D.J. and O.K.C. contributed to protein assay and fluorescence imaging. O.K.C. conceived the ion channel activation experiments and developed these and their analysis together with W.D.J. and J.L. O.K.C. designed and built the optical imaging setup with W.D.J., B.W.D. built the acoustic levitator and contributed to levitator experiments. P.D. and B.W.D. contributed to Multiphysics simulation. W.X. contributed to data analysis. P.R., M.B. and B.M. produced and purified the MscL-G22S as the key reagents. J.L., O.K.C., B.W.D. and D.A.B. wrote the manuscript. All authors discussed the results and commented on the manuscript.

## Competing interests

The authors declare no competing interests.
