## [Peer Review File · Nature Communications]

Building programmable multicompartment artificial cells incorporating remotely activated protein channels using microfluidics and acoustic levitationREVIEWER COMMENTS

Reviewer #1 (Remarks to the Author):

In this manuscript, the authors presented a droplet laboratory platform combining techniques of droplet microfluidics and acoustic levitation to engineer complex emulsion droplets (called artificial cells with distributed cores, ACDC), which can be used as a model system for multicompartment artificial cells.

Both droplet microfluidics and acoustic levitation have been well developed and reported in the past decade, here the authors demonstrated multiple interesting functionalities including contact-free handling of the ACDC with various physical approaches (e.g. air flow control, heating, magnetically polarization, light-initiated reaction). In particular, the authors replicated some well-developed experiments (ref 46) such as Reconstitution of protein channels into the artificial membranes in levitated ACDC droplets, and magnetic control of ion channel gating in levitated ACDC droplet which can mimic the intracellular communication in real biological systems of cells. The results are sound and comprehensive. However, I have some concerns before being published in Nature Communications.

1) Quite some operations have been achieved in the acoustic levitation, however, the patterning and sizes of the droplets in 3D are in mm-cm scale and seems not quite steady, it will bring significant limitations and challenges for high resolution imaging in biological cell scale (~10 μm).

2) In the presented acoustic levitation these artificial cell operations were done in air environment, while natural cells are in aqueous environment. Cells cannot survive for too long in air before drying out. Why the presented operations need to be done in air environments? Can those be done in aqueous environments? If this can be demonstrated, that will be very useful for real biomedical research.

3) The droplet size in this work is 2-3mm with the internal compartment size of ~500 μm , this is about 100 times larger than normal biological cells in diameter which is typically 10-20 μm , the volume difference is one million times. What is the smallest size of droplet that can be handled by the acoustic levitation, including all the operations (such as control of ACDC structural organization, microcentrifuge, heating, polarization, Reconstitution of functional protein channels into the artificial membranes of levitated ACDC droplets, Remote control of ion channel gating in levitated ACDC droplet core networks)? Knowing the limitations of the handling droplet size, and ideally demonstrating some proof of concept, is particularly important for potential biomedical applications.

Overall, this work improves the existing complex emulsion system for more functionalities by optimizing microfluidic design and incorporating the technique of acoustic levitation.

Some minor comments:

1) In the introduction, the authors claim one "problems" they try to solve in this work: "However, to date, few efforts have focused on the permutations of compartments within artificial cells". The permutation of compartments within droplets were achieved by the droplet microfluidics with programable input profile, which is very straightforward/simple and has been demonstrated before. The presented droplet platform looks based on a series of authors' previous works where they already developed the system of complex emulsion droplets in ref. 39&40, including relevant studies on bilayers and magnetically polarization. Please reconsider the statement.

2) The section title "control of ACDC droplet patterning" is confusing, I guess the authors means control of ACDC droplet inner patterning?

3) In the last paragraph of introduction, the authors claim " We observed lipid bilayer formation within the levitated ACDC droplet, where lipid monolayer-coated water cores contact each other (Fig.1E)", however, I can not really "see" the lipid bilayer formation based on the data. Additional data or reference should be provided here for such claim.

4) It is interesting to see the different cores organization in the mother droplet by programming the microfluidic operation. I am wondering whether the acoustic streaming would affect and thus reorganize these cores when the ACDC is trapped in the pressure node? How stable is the core organization?

Reviewer #2 (Remarks to the Author):

In this manuscript the authors present a new platform for the creation, manipulation, control and measurement of complex-emulsion-based supramolecular architectures that they call artificial cells with distributed core droplets (ACDC droplets).

The ACDC droplets are complex emulsion materials constructed from microfluidically-formed, lipid-coated water droplets, here termed "cores", trapped within the nodes (zones) of a vertically orientated acoustic standing wave, generated by a multi-emitter, single-axis acoustic levitator.

A levitated ACDC droplet can be reconfigured, orienting the internal cores by pneumatic rotation, or manipulated by magnetic field application or heat radiation, but radical polymerization can also be induced by light.

I consider this manuscript to be an important step forward in improving high throughput microfluidic techniques for preparing oil droplet architectures, however some concerns and suggestions for improving this work are reported as follows.

General Concerns

Although the platform presented here is quite impressive, however, some doubts arise from the comparison between ACDC droplets and natural cells. The last ones are multicompartiment architecture maintained far from chemical equilibrium by a flux of nutrients from the external environment, while a levitated ACDC droplet appears as an isolated system perturbed only by mechanical, magnetic, thermal and hydrodynamical energy input. Furthermore, the internal architecture of an ACDC droplet consists of delimited single-layer lipid compartments, floating in an oily phase, capable of exchanging molecules only through the reconstituted protein channel in the double-layer surface contact section between adjacent cores. Therefore, no passive transport across the organelle membrane nor diffusion of molecules through the internal milieu, i.e. the cellular cytoplasm, appears to be allowed.

Therefore, in my view, a more critical comparison between ACDC droplets and natural cells and / or a better specification of what the authors really mean by artificial cells would be needed to make the reader truly aware of how far or close we are in building a natural cell with this microfluidic approach.

Additionally, the authors extensively cited articles focusing on microfluidic approaches, however less attention was paid to traditional procedures for preparing giant lipid vesicles (see for instance Walde 2010 *ChemBioChem* 11, 848 – 865) and some recently published results in preparing multicompartimental cell-like architectures (see among others Lee K.Y. et al *Nature Biotechnology* 36, 2018, 530–535, Berhanu S. et al., *Nature Communications* 10, 2019, 1325, Altamura E. et al., *PNAS* 118(7), 2021, e2012170118).

Moreover, some of the authors already published in 2020 (reference 40) a full paper on One-Step, Dual-Material 3D-Printed Microfluidics suitable for the Formation of Polarized, Functional Artificial Cells from Compartmentalized Droplet Networks and Nanomaterials, so it should be better evidenced the difference between these works, in terms of technology improvements/optimizations and experimental results, that could be not so evident for readers not used to microfluidic equipment.

What is the real novelty presented here: acoustic levitation and external air currents to redesign the internal configuration of the cores? Indeed, as early as in the 2020 paper the authors claimed to be able to generate triple emulsions with 3D modelled morphologies and internal droplet populations of different sizes, numbers and hemispherical positions, as well as the ability to create layered and modelled shells.

As the present manuscript is packed with a lot of information, it should make it clearer what is actually new. For example, in my view, the section: ACDC droplet patterning control should not be presented as a result of the new platform but instead as an already acquired capability that has been integrated into the new configuration and this section should be moved to the previous discussion on state of the art of the field.

Specific Concerns

The reconfiguration core network patterns within the ACDC droplet due to the spinning induced by the local air flow seems casual core mixing without any control on the final pattern. On this point, the authors written just a cryptical sentences "Despite the complexity of this structuring mechanism, more robust protocols of core reconfiguration can be established from the mathematical modelling of sphere packing and acoustic assembly [42, 43]. "

It would be better to give more clues about the real possibility or the future strategy to control the reconfiguration of the core pattern rather than the simple detachment and shuffling of the cores that seems to be taking place now.

For example, if it was possible a so precise control on the cores patterning why, in the case of DPhPC-DOPC lipid bilayer experiments, the core assembly have been constructed by adding in situ a DPhPC-coated monolayer core, containing MscL, in the ACDC droplet levitated with a DOPC-based core network? Please give an explanation to the reader because the 3D-Printed Microfluidics apparatus is less suitable to do this task. Or please, explain more clearly if I am wrong.

In the caption of Figure 4, reference is made to a control experiment to verify that no increase in fluorescence is observed in the absence of MscL channels in asymmetric DOPC-DPhPC bilayer networks. Perhaps, the authors refer to Fig S9 in the supporting information (if this is correct, please complete the reference in the caption of Fig. 4).

On the other hand, in the caption of Fig S9 it is not explicitly written that the experiment takes place on asymmetric DOPC - DPhPC nuclei. Therefore, in any case it would be better to report the exact experimental conditions in the caption of Fig. S9 and add, in the SI, the results of the correct control experiment mentioned in the caption of Fig.4, if missing.

Indeed, this control experiment is quite important to demonstrate that spontaneous pores cannot be formed in an asymmetric network of cores due to membrane tension.

Before explaining why mechanosensitive MscL cannot be controlled by magnetically blocked droplets, the authors put a lot of effort and offer mathematical simulations for this purpose, they should at least estimate a success rate in opening / closing the MscL channel with this approach. Indeed, in the diagram of Figure 5F, it seems that the possibility that the nuclei are in a suitable position to be precisely controlled is not that high, perhaps 60%? (Percentage of lighter circles approximated by eyes). In my opinion, this once again places patterning control as a crucial point and how close we are to have an accurate control of cores in this technology.

Although I can understand that the authors are very proud of this promising technology, nevertheless I recommend that, in the conclusion, they also honestly present the weaknesses and discuss the directions for improvement along with the optimization perspectives.

Minor concerns

In the caption of Figure 4, please correct the reference to the control experiment to verify that no increase in fluorescence is observed in the absence of MscL channels in asymmetric DOPC-DPhPC bilayer networks

Fig 5(E) Please rephrase the caption that should comment the curves (ii) and (iii) without specifying again the number of the figure.

Fig S7, Please check the labels of the images that do not correspond to the CaCl₂ concentrations shown on the horizontal axes of the graph.

Reviewer #3 (Remarks to the Author):

The paper describes an intriguing application of droplet microfluidics to create synthetic cells followed by in-air acoustic levitation to provide an environment in which the synthetic cells can be manipulated, modified, and, to an extent, controlled. The use of droplet-microfluidics for artificial cell synthesis has received significant attention in recent years but the combination of that approach with acoustic

levitation is both novel and ambitious. The droplet patterning, an extension of previously published work from this collaboration, provides significant flexibility in creating controlled, patterned cores for the artificial cells. While use of acoustic manipulation has been reported previously in the generation of synthetic cells, this has been in a liquid continuum while here the in-air levitation allows for contact-free suspension and manipulation of the cells under study, and provides a means of reconfiguring the packing within the droplet.

Overall, this has the potential to provide a valuable platform for creating and studying artificial cells and is worthy of publication. The methodology and interpretation of the observations appear sound and the supplementary information provides useful additional detail that should allow for the approach to be reproduced elsewhere.

The description of the heating process could be clarified – presumably the mechanism is convective, but confusingly conduction and irradiation are mentioned in lines 201 & 202 and in the caption for fig 3. It would also be helpful to provide detail on the mechanism by which the introduction of the magnetic field suppresses rotation of the levitated droplets

Reviewer #4 (Remarks to the Author):

The authors describe an innovative system for creating hierarchical droplet assemblies that can be considered as artificial cells. Unique to this work is the combination of microfluidics and acoustic levitation to gain control over multicompart ment droplet assemblies. These tools also enable precise manipulations using magnetic fields for instance. The authors then show that they can reconstitute interesting functions by incorporating membrane proteins such as hemolysin and MScL, and are able to pattern and reconfigure the droplet network in very interesting ways. While the work is highly technical and will be challenging to be adopted by others, I do think the authors have come up with a clever and sophisticated way to create hierarchical assemblies, which the field definitely needs. I only have a few minor suggestions, but recommend publication:

1. I think the authors should include in the manuscript itself a short discussion regarding how they reconstituted the membrane proteins.
2. regarding writing style, I would suggest the authors tone down some of their claims. For instance, referring to their own work as a "milestone" seems unnecessary and remains to be seen.
3. I think the system does have several drawbacks (size of droplets, required instrumentation etc.) the authors should include short discussion about these issues in the discussion.

We thank the referees for their constructive input and feel this has helped improve the submitted work. A point-to-point response to all four reviewer's points is enclosed below:

Reviewer 1:

Comment: Quite some operations have been achieved in the acoustic levitation, however, the patterning and sizes of the droplets in 3D are in mm-cm scale and seems not quite steady, it will bring significant limitations and challenges for high resolution imaging in biological cell scale (~10 μm).

Response: The authors appreciate this comment from the referee. Our acoustic levitator works at 40 KHz, with airborne acoustic waves at that frequency having a wavelength of 8.65 mm at 25 °C, that allows the levitation of object up to ≈ 4 mm (half-wavelength) and significantly smaller. In our experiment, 1~3 mm diameter, microfluidically formed ACDC droplets with controllable size, were efficiently levitated at the acoustic standing wave nodes. According to this, our optical imaging circuit was customised to have sufficient depth of field with long working distance (~11.5 cm,) in order to minimise the influence of equipment proximity on acoustic levitation), to acquire clear bright field and fluorescent images of these levitated ACDC droplets structures. Our imaging setup can record the convective flow of 1 μm microparticles (Fig 1D, Fig 3D-iii) in a levitated droplet, and notably we demonstrate the incorporation of smaller droplets (down to 20 μm) and particulates within our ACDC droplet. These features approach the cellular scale and work by others has demonstrated incorporation of cells within the compartments of droplet networks [1, 2]. Consequently, the reported work could be used with cellular scale components. However, as the referee notes, high resolution imaging cannot be achieved with the current optical setup. However, one can imagine solutions to this to improve imaging resolution at higher magnification, for example operating at a higher ultrasonic frequency to produce smaller nodes allowing closer equipment access for higher magnification, high NA objectives, or the application of shaped ultrasonic fields to compensate for a closer proximity optical objective lens. We note that in the current set-up we have sufficient optical resolution to image ion-flux from membrane ion-channels and that for a number of useful biophysical studies in artificial membrane systems need not be constrained to cellular sized objects. In response to this question, we have addressed this limitation in the discussion section of manuscript, alongside prospects for cell-sized manipulations.

Comment: In the presented acoustic levitation these artificial cell operations were done in air environment, while natural cells are in aqueous environment. Cells cannot survive for too long in air before drying out. Why the presented operations need to be done in air environments? Can those be done in aqueous environments? If this can be demonstrated, that will be very useful for real biomedical research.

Response: In this work, ACDC droplets were isolated following microfluidic manufacture and both manipulated and operated individually in air. The demonstrated operations address the current demand of artificial cell control, e.g. for communication and reconfiguration, that have been discussed in recent literature [3]. This approach affords an expanded repertoire of possible processing and manipulation of artificial cells beyond that achievable by, for example, solely relying on fluidic manipulations within a microfluidic channel following artificial cell production. The air environment provides a scenario in which the droplet processing operations can be conducted and measured in three dimensions with low air friction and spatial restriction, in comparison to the liquid environment. However, there is no reason why these droplets cannot be subsequently be placed in an aqueous environment.

Additionally, the principles could be applied to manipulation in a liquid environment, as has been demonstrated many times, as long as there is sufficient difference in velocity and compressibility of the liquid and the artificial cells. We have previously shown the compatibility of similar microfluidically produced constructs in aqueous environments [4, 5]. We would like to note that whilst cells almost always require an aqueous, rather than an air, environment, artificial cells when used as new soft-matter materials may find applications in alternative environments such as air or oil [1, 6]

Meanwhile, we have demonstrated that new water content can be added to the levitated ACDC droplets as illustrated in Fig 1E, in which the aqueous droplets can be sustained from drying out. In addition, some of the authors have previously demonstrated that acoustic standing waves can be utilised to manipulate particles, living cells, and protocells in an aqueous-based environment [7], providing there is sufficient difference in velocity and compressibility between the liquid and the cellular material. This work indicates the possibility to levitate artificial cells in an all aqueous environment for the replication of these air-based operations. This would require further development of the underpinning acoustics and is beyond the scope of this work. However, these limitations and considerations have been added to the discussion, where we also highlight the possibility of considering the ACDC droplet a chassis which when levitated provides a water environment within the internal cores for cells or other aqueous-compatible biochemistry.

Comment: The droplet size in this work is 2-3mm with the internal compartment size of ~500µm, this is about 100 times larger than normal biological cells in diameter which is typically 10-20µm, the volume difference is one million times. What is the smallest normal biological cells in diameter which is typically 10-20µm, the volume difference is one million times. What is the smallest size of droplet that can be handled by the acoustic levitation, including all the operations (such as control of ACDC structural organization, microcentrifuge, heating, polarization, Reconstitution of functional protein channels into the artificial membranes of levitated ACDC droplets, Remote control of ion channel gating in levitated ACDC droplet core networks)? Knowing the limitations of the handling droplet size, and ideally demonstrating some proof of concept, is particularly important for potential biomedical applications.

Response: Artificial cell and membrane models have already attracted significant interest across a wide-range of applications at comparable length scales to the artificial cells manipulated in our study, for example for ion channel screening [8], the development of 3D printed tissues [9], DNA sequencing [10], drug membrane permeability screening [11] and the study of fundamental membrane properties [12]. This is in addition to technological demonstrations of artificial cells as advanced functional materials, e.g. artificial eye [13], biological logic circuits [14] and as materials that can interface with living cells and tissues [15]. As such we believe there is plenty of scientific scope for application without necessarily miniaturising to the biological cellular scale. Having said that, within the field these droplet interface bilayers (DIBs - the droplet membrane system that underpin the membrane structural components of our ACDC artificial cells) researchers (ourselves and others) have miniaturised DIBs with the successful incorporation of membrane spanning channels [16] at these scales. Separately, acoustic levitation and manipulation of biological cells has been demonstrated [17] and consequently in principle both approaches can be combined at this smaller length scale. We note that in our reported work here, whilst the overall construct size is 1+ mm, individual droplets within the internal droplet network are typically on the order of ~100 µm and may be made as small as 20 µm, coming within the order of magnitude of the cellular length scale. In such cases the larger droplet can be viewed as a carrier for the smaller droplet interface bilayer network. We have operated at the reported length scale as this is fairly typical within the DIB field.

Our levitator can levitate smaller droplets down to ~100 µm in air. At such scales, the levitated droplets are more susceptible to external perturbation (e.g. ambient air currents) and the asymmetry of the system (e.g. transducer array setup), due to the acoustic streaming effect. So if desired, operation at this scale would benefit from encasing the experimental system in a cabinet, or similar, to avoid perturbation from air currents within the room, as well as developing new acoustic device for precision acoustic manipulation of cell-sized objects at microscale resolution. In our experiments the levitator experimental setup is simply housed on a laboratory bench and so conditions are not optimised for this purpose.

In previous publications from authors within the collaborative team [18], similar acoustic levitators have been used to trap aerosol spray to “grow” water droplets at the acoustic standing wave nodes in air. Hence, in principle, the presented acoustic levitation and operations can manipulate cell-sized droplets in air, that can be gathered and then controlled and measured collectively with further droplet processing methods and associated imaging equipment development. These limitations and considerations have been addressed in the discussion.

Meanwhile, natural bacteria and single cell organism can grow to millimetre to centimetre sizes [19]. For bottom-up synthetic biology research, the size of artificial cell models ranges from hundreds of nanometres to a few millimetres, depending on the nature of the research objectives [20]. Our ACDC droplet model focused on the control of multi-compartment architectures (and so could also be viewed as artificial tissues made up of multiple artificial cells [9]) and the study of inter-compartment communication via reconstituted transmembrane protein in the assembled droplet interface bilayer membranes. These topics are of significant interest in bottom-up artificial cell creation.

Some minor comments:

Comment: In the introduction, the authors claim one “problems” they try to solve in this work: “However, to date, few efforts have focused on the permutations of compartments within artificial cells”. The permutation of compartments within droplets were achieved by the droplet microfluidics with programable input profile, which is very straightforward/simple and has been demonstrated before. The presented droplet platform looks based on a series of authors’ previous works where they already developed the system of complex emulsion droplets in ref. 39&40, including relevant studies on bilayers and magnetically polarization. Please reconsider the statement.

Response: We thank the referee for this observation and are very happy to amend and clarify the statement in the introduction. We have revised this to state “However, few efforts have focused on the permutation of compartments within artificial cell, and the subsequent processing of these architectures and their functions”. This statement is more specific for engineering artificial cells, rather than a more general statement on permutation of droplets with multiphase microfluidics. –Within the context of microfluidic engineering of artificial cells, the current state-of-the-art is not routinely controlling the contents AND organisation of internal compartments [21,

22], instead usually favouring manual or individual construction for droplet networks to achieve precise arrangement of droplets of different content. Whilst we have demonstrated advancement of the microfluidic construction previously, as the referee notes, with use of 3D fabrication space to localise droplets or encapsulated material to defined hemispheres of hierarchical emulsion droplets, in the work reported here we expand significantly with post-process manipulation (including reconfiguration of the internal droplet organisation) and functional activation of membrane proteins.

Comment: The section title “control of ACDC droplet patterning” is confusing, I guess the authors means control of ACDC droplet inner patterning?

Response: We have removed the section title of “ACDC droplet patterning control”, and modified the context and paragraphs with a new section title “Acoustic levitation of ACDC droplet”. In this way, the novelty present in the manuscript will be better highlighted.

Comment: In the last paragraph of introduction, the authors claim” We observed lipid bilayer formation within the levitated ACDC droplet, where lipid monolayer-coated water cores contact each other (Fig.1E)”, however, I cannot really “see” the lipid bilayer formation based on the data. Additional data or reference should be provided here for such claim.

Response: Given the very thin nature of lipid bilayer (~10 nm thickness), the referee is correct that they are consequently indirectly, rather than directly, observed. We evidence their presence by robust and established approaches in the droplet interface bilayer research community and have added further references to support. Apologies if this was not so obvious initially. Based on experience, those in the field will confirm that droplet interface bilayers (DIBs) form where two aqueous bodies contact within an oil environment in the presence of dissolved lipid. This lipid, being amphiphilic, self-assembles at the water-oil interface reducing interfacial tension. Where two such lipid monolayers meet a lipid bilayer is formed. This has recently become a popular and now established method for forming artificial lipid bilayer [23] – in the literature bilayers are typically evidenced by one of; electrophysiology membrane capacitance measurement [12]; or evidencing successful reconstitution of membrane proteins that crucially require a membrane environment to function [24]. A stable lipid bilayer formed between two contacting aqueous droplets prevents mixing of the components either side of the lipid bilayer and visually manifests as a flat surface (interface) observed at the boundary of the two juxtaposed droplets – hence termed droplet interface bilayers.

In the work reported here we experimentally evidence the formation of lipid bilayers in several ways: (1) we observe no mixing of droplet contents between two adjoining droplets (i.e. no calcium – fluo8 mixing is observed with these species remaining separated by the lipid bilayer – this evidences a non-permeable barrier between the contacting droplets. (2) the incorporation of aHL membrane protein facilitates the flux of calcium ions across the lipid bilayer through the protein pore eliciting a fluorescence response when subsequently binding to fluo-8 in the recipient droplet. This has become an established approach for evidencing the presence of lipid bilayers [14]. (3) Further, in the reported work, we incorporate the membrane spanning ion-channel MscL and evidence calcium flux through this channel under conditions where we expect the channel to be activated (opened) (i.e. asymmetric lipid bilayer, addition of LPC and induction of membrane tension) – in the absence of the protein none of these stimuli give rise to ion-flux, evidencing the ion flux is occurring through the MscL channel embedded in a lipid bilayer. These results are presented in the manuscript, and we are confident those working in the field of droplet interface bilayers will view this as convincing evidence of bilayer formation. However, for those less familiar, new SI figures (Fig. S4) and further reference have been added to provide further information of droplet interface bilayers formation for the reader. We thank the referee for pointing out that this may not be apparent to those approaching this work without this specific familiarity and we hope the added references and figure aid understanding.

Comment: It is interesting to see the different cores organization in the mother droplet by programming the microfluidic operation. I am wondering whether the acoustic streaming would affect and thus reorganize these cores when the ACDC is trapped in the pressure node? How stable is the core organization?

Response: The adhesive force between contacting aqueous droplets can be indirectly measured by the contact angle at the droplet interface [1]. Notably, the adhesive van der Waals forces of the lipid bilayer are sufficient to deform the droplets from a spherical shape to drive a flat adherent surface between two (or more) contacting droplets. Consequently, with a small internal droplet network we observe the whole droplet network tending to move as a single unit, as a result of the acoustic streaming induced convections within the levitated, host, oil droplet. Our MscL experiments show that the rotation and subtle reorganisation of the droplet network is sufficient to dissipate any induced forces without activating the membrane tension sensitive MscL protein, and without observing any reorganisation of the connectivity of the network. We are however able to reorganise the droplet network by application of greater forces using the demonstrated air flow technique to spin the ACDC construct rapidly (>1500 rpm) which can provide sufficient force to detach the lipid bilayers of the droplet network and

facilitate droplet network reorganisation based on factors including droplet size and density. In our opinion, the acoustic streaming may influence the reorganisation of cores when they are detached. Such dynamics will be investigated in the future, we anticipate affording further control over droplet network reorganisation.

Reviewer 2

Comment: Although the platform presented here is quite impressive, however, some doubts arise from the comparison between ACDC droplets and natural cells. The last ones are multicompartment architecture maintained far from chemical equilibrium by a flux of nutrients from the external environment, while a levitated ACDC droplet appears as an isolated system perturbed only by mechanical, magnetic, thermal and hydrodynamical energy input.

Furthermore, the internal architecture of an ACDC droplet consists of delimited single-layer lipid compartments, floating in an oily phase, capable of exchanging molecules only through the reconstituted protein channel in the double-layer surface contact section between adjacent cores. Therefore, no passive transport across the organelle membrane nor diffusion of molecules through the internal milieu, i.e. the cellular cytoplasm, appears to be allowed.

Therefore, in my view, a more critical comparison between ACDC droplets and natural cells and / or a better specification of what the authors really mean by artificial cells would be needed to make the reader truly aware of how far or close we are in building a natural cell with this microfluidic approach.

Response: In the discussion section, we have added a description of the ACDC droplet as a potential multi-compartment model (chassis), which can be employed to impart cellular-like functions within an artificial cell system. Current bottom-up artificial cell models are far less complicated than biological cells, as the reviewer notes, however, by introducing conceptually analogous processes to those employed by biological cells, soft matter materials with increasingly sophisticated properties and function can be engineered. This is highlighted by the advances in properties demonstrated in multi-compartment artificial systems compared to single compartment droplets or vesicles, where even small networks of droplet can adopt interesting emergent function [9, 22, 23, 25]. We describe in the introductory section, that one of the major challenges is to replicate or mimic a greater range of cell functionalities within artificial cell systems. This requires precision engineering methods to control chemical organisation and compartmentalisation to generate more sophisticated artificial cell materials, as an increasingly vivid counterpart to living cells. Our work aims to address this objective not only by employing precision droplet formation and chemical encapsulations using microfluidic approaches, but also through the post-processing of droplet structures to alter the characteristics of the artificial cell *in situ*. With such capability, we have investigated how the building blocks of artificial cells, e.g. compartments, membranes and proteins, react in different scenarios towards engineering increasingly complex artificial cell behaviours or properties. For example, the introduction here of the ability to control protein function to both activate and deactivate membrane protein activity pushes beyond the current standard of single time activation.

We appreciate the reviewer's comment that this is still some distance from a biological cell. Within this research area, ourselves and others are at the stage of seeking to create architectures capable of similar or analogous processes of those of biological cells (e.g. affording control of chemical reactions in space and time within soft-matter materials), even if achieved via different mechanisms than those employed by biological cells. It is noteworthy that whilst lipid vesicles have provided a fruitful platform of much study as an artificial cell chassis or synthetic cell model, that the relatively simple conceptual advance to multiple (rather than singular) compartments, even when joined in simple networks, are in fact able to demonstrate a range of interesting emergent properties [26], much like how biological tissues display more sophisticated function than individual cells. Consequently, compartmentalised artificial architectures with controlled communication between compartments affords the prospect of enabling significantly enhanced functional utility compared to single compartment structures. As such, rather than create a carbon copy of a biological cell, we rather seek to understand and demonstrate the possibilities afforded by soft-matter engineering that is inspired by biology. In this context the ability to create networked compartments, separated by lipid bilayers, with the incorporation of functional membrane proteins that can be externally controlled, enables and advances the controlled chemical communication between these compartments for the attainment of higher-order function that is not possible in an open pot or single compartment systems.

Comment: Additionally, the authors extensively cited articles focusing on microfluidic approaches, however less attention was paid to traditional procedures for preparing giant lipid vesicles (see for instance Walde 2010 ChemBioChem 11, 848 – 865) and some recently published results in preparing multicompartmental cell-like architectures (see among others Lee K.Y. et al Nature Biotechnology 36, 2018, 530–535, Berhanu S. et al., Nature Communications 10, 2019, 1325, Altamura E. et al., PNAS 118(7), 2021, e2012170118).

Response: It was not our intention to overlook these systems and indeed giant lipid vesicle are a well-studied protocell model that have been widely used for the encapsulation of biochemical or metabolic reactions. We have modified our introduction to reference these works in lipid vesicles and ATP productions in protocells.

Comment: Moreover, some of the authors already published in 2020 (reference 40) a full paper on One-Step, Dual-Material 3D-Printed Microfluidics suitable for the Formation of Polarized, Functional Artificial Cells from Compartmentalized Droplet Networks and Nanomaterials, so it should be better evidenced the difference between these works, in terms of technology improvements/optimizations and experimental results, that could be not so evident for readers not used to microfluidic equipment.

What is the real novelty presented here: acoustic levitation and external air currents to redesign the internal configuration of the cores? Indeed, as early as in the 2020 paper the authors claimed to be able to generate triple emulsions with 3D modelled morphologies and internal droplet populations of different sizes, numbers and hemispherical positions, as well as the ability to create layered and modelled shells.

Response: Our previous work [4, 5] focused on the microfluidic approach to generate complex emulsions for the creation of artificial cell chassis. In this regard, the major advances were the adoption of dual-material printing to create complex emulsions without the need of surface modification, and the use of truly 3D-fluidic architectures to enable spatial manipulations and patterning of droplets not possible in more conventional 2.5D microfluidic geometries. The submitted manuscript here reports an acoustic approach to levitate and post process microfluidically formed complex emulsion droplets for remote and contactless manipulation and control of artificial cell behaviours – e.g. membrane protein activation/de-activation. We believe this is a highly significant advance that synergises with the microfluidic creation of complex emulsions. In the course of this work, we also increased the sophistication of the microfluidics used to generate complex emulsions, introducing advances in printing fidelity that enable to production of smaller droplets (~20 μm) and the establishment of fluid flow control to create more sophisticated droplet sequences. However, we acknowledge that these advances are more technical and specialist in nature. We thank the reviewer for raising this question and have sought to highlight the novelty more clearly in our revised manuscript, for example stating, “In this work, we focus on the processing methods for microfluidically formed ACDC droplets, specifically on the organisation and control of compartment networks in three dimensions.”, alongside rearranging the contents and paragraphs for emphasis and clarity. We have re-titled the first results to focus on the novelty and reconfigured the closing paragraph of this section to articulate the microfluidic advances more specifically more clearly.

Comment: As the present manuscript is packed with a lot of information, it should make it clearer what is actually new. For example, in my view, the section: ACDC droplet patterning control should not be presented as a result of the new platform but instead as an already acquired capability that has been integrated into the new configuration and this section should be moved to the previous discussion on state of the art of the field.

Response: We thank the referee for this comment which together with the previous comment has encouraged us to reflect on how our work was presented. To aid with clarity and communication of the most important and novel features of the work, we have removed the section title of “ACDC droplet patterning control” and modified the context and paragraphs with a new section title “acoustic levitation of ACDC droplets in air”. We have more clearly articulated near the start of this section how this work builds upon (and differs) from our earlier reported work. In this way, we believe the advances and novelty present in this work is better highlighted.

Specific Concerns

Comment: The reconfiguration core network patterns within the ACDC droplet due to the spinning induced by the local air flow seems casual core mixing without any control on the final pattern. On this point, the authors written just a cryptical sentences “Despite the complexity of this structuring mechanism, more robust protocols of core reconfiguration can be established from the mathematical modelling of sphere packing and acoustic assembly [42, 43]. “It would be better to give more clues about the real possibility or the future strategy to control the reconfiguration of the core pattern rather than the simple detachment and shuffling of the cores that seems to be taking place now.

Response: In current work, the reconfiguration of internal cores uses the mechanical approach shown in Fig 3B, that airflow induced droplet spinning and the centrifugal force, causes the detachment of droplets (core) from the internal network. The adhesive force of the lipid bilayers (discussed above) has to be overcome for droplet detachment and the core network disassembly. The magnitude of this adhesive force scales with bilayer area, and therefore is linked to droplet size and droplet-droplet contact area. Once the air flow is ceased or slowed, droplet spinning is slowed down, and so the separated cores return to a central position, to contact with each other, and form a new droplet interface bilayer network. Consequently, the forces generated on individual droplets in disassembly and reassembly are heavily influenced by the droplet size and density (in addition to current droplet configuration). These physical parameters thus offer variables for control and ultimately predictability of the reconfiguration. Here, we use control of droplet size (via the upstream microfluidics) and the aqueous density of individual droplet using sucrose to affect these variables. The initial droplet packing arrangement is determined by the microfluidically produced droplet sequence, and the air flow speed used to control the rate of spinning. In the reported example, two droplet populations (yellow and pink) of different sizes and density are reconfigured

based on this approach and thus the resultant arrangements are not a consequence of random shuffling. In this manuscript we have not comprehensively studied this physical phenomenon nor the additional influence of acoustic pressure variation within the levitated ACDC droplet to place limits on operation or conduct more extensive operations. However, this is a promising approach to control the reconfiguration of artificial cell compartmentalisation, as well as more general microfluidic emulsification processes, which we believe will be of interest to both communities and worthy of reporting. With even simple droplet networks demonstrating emergent behaviours dependent upon the sequence of droplet connectivity [9, 27], the ability to dynamically reconfigure a pallet of droplet components, even if restricted to a limited number of configurations, holds interesting promise for functional plasticity of such materials. We would like to thank the reviewer for raising this query and believe the modified manuscript is now significantly clearer in how the droplet reconfiguration is achieved on the basis of droplet size and density, which was not well articulated previously. We have rephrased the text of this section “On cessation of air flow, the spinning of the ACDC droplet slows down, and the separated cores reassemble, again forming lipid bilayers between contacting cores. New core networks may be assembled in this way, *with the volumes and densities of the cores, together with air flow rate, determining the release order and reassembly.* This bilayer disassembly-assembly process can therefore be used to reconfigure the core network patterns within the ACDC droplet, changing the packing order. For instance, Fig 3B-iv shows reconfiguration by this method. *By controlling the fluidic input profile and sucrose concentration in two aqueous input flows, an internal core network comprised of different size and density, green and red droplets were created, that creates different patterns of core-core connectivity following spin-based separation and reassembly.*” We have also added further discussion in the manuscript discussion/conclusion section to provide comment on prospects for further refinement with the application of locally modulated acoustic fields.

Comment: For example, if it was possible a so precise control on the cores patterning why, in the case of DPhPC-DOPC lipid bilayer experiments, the core assembly have been constructed by adding in situ a DPhPC-coated monolayer core, containing MscL, in the ACDC droplet levitated with a DOPC-based core network? Please give an explanation to the reader because the 3D-Printed Microfluidics apparatus is less suitable to do this task. Or please, explain more clearly if I am wrong.

Response: In our reported experiment there were several contributing factors behind the manual addition of the protein containing droplet in the experiment referred above; principally, this method allowed us to reliably make an asymmetric lipid bilayer whilst retaining the ‘lipid-out’ approach of using lipid dissolved in the oil phase. Since all internal cores share a common oil environment containing the dissolved lipid, the incubation of a separate droplet in an oil with a different dissolved lipid (e.g. DOPC vs DPhPC) and subsequent droplet transfer allows for easy formation of asymmetric lipid bilayers since the lipid that self-assembles to form the lipid monolayer surrounding the added droplet is different (DPhPC) to that of the prior-encapsulated droplets (DOPC) and thus an asymmetric lipid bilayer is formed where these contact. By using this lipid-out method we maximise the asymmetry of the bilayer leaflets of the formed membrane due to the minimised mixing of DOPC and DPhPC lipids compared to if using a single microfluidic device. Our manual operation can be replicated with two parallel 3D-printed microfluidic devices or two droplet forming and incubation channels, with one specified for DPhPC-coated droplet generation and the other is for DOPC-coated droplet generation, to prevent cross contamination of lipids during lipid monolayer assembly. In principle, the lipid-in method to build customised droplet network is also achievable in a microfluidic approach [28]. We really appreciate that the reviewer pointed this out, in response we have added a new SI figure to explain the this.

We further followed this strategy used to form asymmetric membrane with reconstituted protein pores also based on the considerations of preservation of protein, especially for the custom produced and purified mutant MscL-G22S. For the protein containing buffer we would typically use ~ 10 μ L per day by adopting this approach, and could readily keep protein stocks cold until immediately prior to use. Whilst maintenance of microfluidic reservoir temperature is possible with temperature control solutions, our fluid delivery system would require a minimum volume of ~200 μ L with careful dispersion. Thus, whilst solutions for low volume fluid delivery into microfluidic systems exist and can be integrated into our fluidic delivery system to minimise this volume, for practical considerations this was outside the scope of this work, with us instead using coloured cores to exemplify the core organisation experiments. Additionally, we also found it interesting that the levitating ACDC construct was sufficiently stable to accommodate *in situ* pipetted additions without being displaced.

Comment: In the caption of Figure 4, reference is made to a control experiment to verify that no increase in fluorescence is observed in the absence of MscL channels in asymmetric DOPC-DPhPC bilayer networks. Perhaps, the authors refer to Fig S9 in the supporting information (if this is correct, please complete the reference in the caption of Fig. 4).

On the other hand, in the caption of Fig S9 it is not explicitly written that the experiment takes place on asymmetric DOPC - DPhPC nuclei. Therefore, in any case it would be better to report the exact experimental conditions in

the caption of Fig. S9 and add, in the SI, the results of the correct control experiment mentioned in the caption of Fig.4, if missing.

Indeed, this control experiment is quite important to demonstrate that spontaneous pores cannot be formed in an asymmetric network of cores due to membrane tension.

Response: We thank the referee for pointing this out. Previous Fig S9 (now Fig S13) is an example of a DPhPC symmetric bilayer, and we highlighted this condition in the caption. We have also conducted control experiments of the asymmetric (DOPC-DPhPC) bilayer without MscL to address this comment. These are now incorporated as Fig S11, the results indicates that without MscL, asymmetric DOPC-DPhPC lipid membranes did not form spontaneous pores in acoustic levitated ACDC droplets, with either locked or non-locked magnetic operation (i.e. no ion-flux observed fluorescence). To summarise for the MscL experiments, in both symmetric and asymmetric membranes without MscL we observe no evidence of pore formation (ion-flux) in acoustically levitated ACDC droplets, either with or without the locked magnetic operation. Only in the presence of MscL and either an asymmetric membrane to induce tension, or where a symmetric membrane is used the application of the magnetic-lock operation, is evidence of active pore function and ion-flux observed.

Comment: Before explaining why mechanosensitive MscL cannot be controlled by magnetically blocked droplets, the authors put a lot of effort and offer mathematical simulations for this purpose, they should at least estimate a success rate in opening / closing the MscL channel with this approach. Indeed, in the diagram of Figure 5F, it seems that the possibility that the nuclei are in a suitable position to be precisely controlled is not that high, perhaps 60%? (Percentage of lighter circles approximated by eyes). In my opinion, this once again places patterning control as a crucial point and how close we are to have an accurate control of cores in this technology.

Although I can understand that the authors are very proud of this promising technology, nevertheless I recommend that, in the conclusion, they also honestly present the weaknesses and discuss the directions for improvement along with the optimization perspectives.

Response: We thank the reviewer for this comment as this has prompted us to conduct additional work to estimate force generation as well as reflect upon our original presentation and articulation of this section of the manuscript. MscL channels respond to membrane tension (10–12 mN/m) [29, 30]. In thinned bilayers, MscL channel activates at lower pressures and intermediate states are stabilized. MscL-G22S, which is used in this work is not spontaneously active but has a lower activation threshold than wild-type MscL [31]. For this work, the absence of spontaneous channel gating affords a defined off-state in the absence of activating forces. We thank the reviewer for this interesting question on the extent of activation in the reported system and have expanded our modelling work to assess this. The results of this indicate that levitating an ACDC droplet requires approximately 2200-2800 Pa generated by acoustic standing waves (Fig S12). The applied acoustic field generates acoustic streaming flows within the levitated droplet system. These flows generate fluidic shear on the lipid bilayers. Our modelling in the initially submitted manuscript demonstrated that this acoustic flow velocity is greatest below the mid-plane in vertical positioning and near the ACDC droplet perimeter in lateral position (Fig 5G) and thus in these regions we expect greater shear on the lipid bilayers. We observed that this modelling corresponded remarkably well with the experimentally observed extent of MscL activation (Fig 5F). The intention in of the experiments of Fig 5F and the corresponding modelling of Fig 5G is to demonstrate that the MscL channel activation is predictable and based upon the droplet position within the ACDC capsule, which determines the amount of sheer stress experienced. As such we have deliberately sought to experimentally explore the positional space of the location of the protein containing droplet.

In further modelling (Fig S12 employing representative fluid properties of water, hexadecane, and air), we modelled and evaluated the fluidic shear tension, which is induced by the acoustic radiation force and associated acoustic streaming effect, exerted on an example of droplet network (as shown in Figure 5G). The tension is calculated by the integral of the multiplication of fluidic shear rate and the fluid viscosity using COMSOL Multiphysics software (version 5.6). The value of this tension is within the range of 8–16 mN/m, which is comparable to the tension requirement for actuation of MscL gating. This generated tension is related to the fluid properties and the fluid streaming velocity within the droplet compartment and so may be modulated with control of acoustic pressure and droplet contents, as well as droplet position. We have expanded the discussion to incorporate these as current limitations and future prospects, also noting that “*shaping the acoustic field with consideration of the ACDC droplet structure (e.g. by independently addressing solenoids of the multi-emitter source) it should be possible to target activation to other regions.*”

The intention of the experiments of Fig 5F and the corresponding modelling of Fig5G is to demonstrate that the MscL channel activation is predictable and based upon the droplet position within the ACDC capsule, which determines the amount of shear stress experienced. Under the magnetic operating conditions, we observe greatest MscL activity corresponding to ~88% of the activity observed in asymmetric bilayer conditions. Whilst as shown in Fig 5F, the MscL activity has a similar pattern to the simulated fluid dynamics profile (Fig 5G), indicating the

possibility that the MscL activity in our model, depends upon the local fluidic shear tension, across all explored protein containing core locations. This data-driven interpretation indicates that the presented MscL gating control, can be optimised with better organisation of core network (e.g. core size, position and connection), incorporating predictive modelling, as well as tuning other input parameters, such as acoustic radiation force, surrounding temperature and fluid properties. This inference emphasises that the precision control of chemical compartmentalisation, underpinned by microengineering capabilities, is important for developing functional artificial cell.

Based on above, we have modified the manuscript context, description and discussion concerning these MscL activation experiments, we have more clearly articulated the current limitations of this approach whilst in the conclusions suggesting prospects to build upon this. We have added new SI figure of the new modelling results estimating shear tension.

Minor concerns

Comment: In the caption of Figure 4, please correct the reference to the control experiment to verify that no increase in fluorescence is observed in the absence of MscL channels in asymmetric DOPC-DPhPC bilayer networks

Fig 5(E) Please rephrase the caption that should comment the curves (ii) and (iii) without specifying again the number of the figure.

Fig S7, Please check the labels of the images that do not correspond to the CaCl₂ concentrations shown on the horizontal axes of the graph.

Response: We have modified the context and legend based on the reviewer's comments and conducted the additional control experiments.

Reviewer 3

Comment: The description of the heating process could be clarified – presumably the mechanism is convective, but confusingly conduction and irradiation are mentioned in lines 201 & 202 and in the caption for fig 3

Response: We thank the reviewer for pointing this out. We modified the description of heating process for consistency in the context and the fig 3 caption.

Comment: It would also be helpful to provide detail on the mechanism by which the introduction of the magnetic field suppresses rotation of the levitated droplets

Response: The incorporation of magnetic particles within aqueous droplets has been used to manipulate droplet interface bilayer networks in an oil environment, with the magnetic force providing sufficient force to move the droplets without breaking the lipid bilayers [32]. Here, a similar principle is employed, whereby the incorporation of hydrophilic magnetic particles within one of the cores of the ACDC capsule (as shown in Fig3) means the application of a magnetic field can exert a force on this core of sufficient magnitude to orientate the whole ACDC droplet, without pulling the ACDC droplet from the levitator field. In the absence of the magnetic field and/or encapsulated magnetic particles, the whole ACDC droplet construct is free to rotate or move within the acoustic pressure node to dissipate any forces manifest from local air-currents or momentary asymmetry of the acoustic pressure.

Reviewer 4

Comment: I think the authors should include in the manuscript itself a short discussion regarding how they reconstituted the membrane proteins.

Response: we modified the context and add a new SI figure (Fig S8) to explain the reconstitution of membrane proteins and also added additional reference to wider work on the reconstitution of membrane proteins into droplet interface bilayers.

Comment: regarding writing style, I would suggest the authors tone down some of their claims. For instance, referring to their own work as a "milestone" seems unnecessary and remains to be seen.

Response: Thank you for this helpful feedback, we have modified the manuscript writing in response to this comment.

Comment: I think the system does have several drawbacks (size of droplets, required instrumentation etc.) the authors should include short discussion about these issues in the discussion.

Response: We have expanded the discussion section of the manuscript to cover limitations and drawbacks and specifically those mentioned by the referee here. We have added a section in the wrapping up discussion around the size of droplets used here and the opportunities for miniaturisation.

Reference:

1. Villar, G., Heron, A. J., & Bayley, H. Formation of droplet networks that function in aqueous environments. *Nature nanotechnology* **6**, 803-808 (2011).
2. Xavier, C. I. Microfluidic generation of encapsulated droplet interface bilayer networks (multisomes) and their use as cell-like reactors. *Chemical Communications* **52**, 5961-5964 (2016).
3. Smith, J. M., Chowdhry, R., & Booth, M. J. Controlling synthetic cell-cell communication. *Frontiers in Molecular Biosciences* **8**, 809945 (2021).
4. Baxani, D. K. et al. Bilayer networks within a hydrogel shell: a robust chassis for artificial cells and a platform for membrane studies. *Angewandte Chemie International Edition* **55**, 14240-14245 (2016).
5. Li, J. et al. Formation of Polarized, Functional Artificial Cells from Compartmentalized Droplet Networks and Nanomaterials, Using One-Step, Dual-Material 3D-Printed Microfluidics. *Advanced Science* **7**, 1901719 (2020).
6. Boreyko, J. B., Polizos, G., Datskos, P. G., Sarles, S. A., & Collier, C. P. Air-stable droplet interface bilayers on oil-infused surfaces. *Proceedings of the National Academy of Sciences* **111**, 7588-7593 (2014).
7. Wang, X. et al. Chemical information exchange in organized protocells and natural cell assemblies with controllable spatial positions. *Small* **16**, 1906394 (2020).
8. Sapra KT, Bayley H. Lipid-coated hydrogel shapes as components of electrical circuits and mechanical devices. *Scientific reports* **2**, 1-9 (2012).
9. Villar, G., Graham, A. D., & Bayley, H. A tissue-like printed material. *Science* **340**, 48-52 (2013).
10. Howorka, S., Cheley, S., & Bayley, H. Sequence-specific detection of individual DNA strands using engineered nanopores. *Nature biotechnology* **19**, 636-639 (2001).

11. Seddon, A. M., et al. Drug interactions with lipid membranes. *Chemical Society Reviews* **38**, 2509-2519 (2009).
12. Gross, L. C., Heron, A. J., Baca, S. C., & Wallace, M. I. Determining membrane capacitance by dynamic control of droplet interface bilayer area. *Langmuir* **27**, 14335-14342 (2011).
13. Hoskin CE, Schild VR, Vinals J, Bayley H. Parallel transmission in a synthetic nerve. *Nature Chemistry* Apr 21:1-8 (2022).
14. Huang, S., Romero-Ruiz, M., Castell, O. K., Bayley, H., & Wallace, M. I. High-throughput optical sensing of nucleic acids in a nanopore array. *Nature nanotechnology* **10**, 986-991 (2015).
15. Zhou, L., et al. Lipid-bilayer-supported 3D printing of human cerebral cortex cells reveals developmental interactions. *Advanced Materials* **32**, 2002183 (2020).
16. Booth, M. J., Schild, V. R., Graham, A. D., Olof, S. N., & Bayley, H. Light-activated communication in synthetic tissues. *Science Advances* **2**, e1600056 (2016).
17. Wang, X., et al. Chemical information exchange in organized protocells and natural cell assemblies with controllable spatial positions. *Small* **16**, 1906394 (2020).
18. Marzo, A., Barnes, A., & Drinkwater, B. W. TinyLev: A multi-emitter single-axis acoustic levitator. *Review of Scientific Instruments* **88**, 085105 (2017)..
19. Jean-Marie Volland et al. A centimeter-long bacterium with DNA compartmentalized in membrane-bound organelles bioRxiv 2022.02.16.480423; doi: <https://doi.org/10.1101/2022.02.16.480423>
20. Chang, T. M. S. Artificial cell bioencapsulation in macro, micro, nano, and molecular dimensions: keynote lecture. *Artificial cells, blood substitutes, and biotechnology* **32**, 1-23 (2004).
21. Supramaniam, P., Ces, O., & Salehi-Reyhani, A. Microfluidics for artificial life: techniques for bottom-up synthetic biology. *Micromachines* **10**, 299 (2019).
22. Booth, M. J., Schild, V. R., Downs, F. G., & Bayley, H. Functional aqueous droplet networks. *Molecular BioSystems* **13**, 1658-1691 (2017).
23. Castell, O. K., Berridge, J., & Wallace, M. I. Quantification of membrane protein inhibition by optical ion flux in a droplet interface bilayer array. *Angewandte Chemie International Edition* **51**, 3134-3138 (2012).
24. Thompson, J. R., Cronin, B., Bayley, H., & Wallace, M. I. Rapid assembly of a multimeric membrane protein pore. *Biophysical journal* **101**, 2679-2683 (2011).
25. Hindley, J. W., Law, R. V., & Ces, O. Membrane functionalization in artificial cell engineering. *SN Applied Sciences* **2**, 1-10 (2020).
26. Downs, F. G., et al. Multi-responsive hydrogel structures from patterned droplet networks. *Nature chemistry* **12**, 363-371 (2020).
27. Maglia, G., et al. Droplet networks with incorporated protein diodes show collective properties. *Nature nanotechnology* **4**, 437-440 (2009).
28. Hwang, W. L., Chen, M., Cronin, B., Holden, M. A., & Bayley, H. Asymmetric droplet interface bilayers. *Journal of the American Chemical Society* **130**, 5878-5879 (2008).
29. Berrier, C., Besnard, M., Ajouz, B., Coulombe, A., & Ghazi, A. Multiple mechanosensitive ion channels from *Escherichia coli*, activated at different thresholds of applied pressure. *The Journal of membrane biology* **151**, 175-187 (1996).
30. Sukharev, S. I., Sigurdson, W. J., Kung, C., & Sachs, F. Energetic and spatial parameters for gating of the bacterial large conductance mechanosensitive channel, MscL. *The Journal of general physiology* **113**, 525-540 (1999).
31. Rosholm, K. R., et al. Activation of the mechanosensitive ion channel MscL by mechanical stimulation of supported Droplet-Hydrogel bilayers. *Scientific reports* **7**, 1-10 (2017).
32. Wauer, T., et al. Construction and manipulation of functional three-dimensional droplet networks. *ACS nano* **8**, 771-779 (2014).

REVIEWERS' COMMENTS

Reviewer #1 (Remarks to the Author):

I can see that the authors took a great effort to address the review comments, and they did address most of them. However, the main concern as below was not addressed. If the droplets are not stable in the system, it will significantly limit its application where imaging is needed.

“Quite some operations have been achieved in the acoustic levitation, however, the patterning and sizes of the droplets in 3D are in mm-cm scale and seems not quite steady, it will bring significant limitations and challenges for high resolution imaging in biological cell scale (~10 μm).”

It is still not quite clear where the limitation is regarding the size of the droplet that can be handled by the acoustic levitation, with the operations in the manuscript (such as control of ACDC structural organization, microcentrifuge, heating, polarization, Reconstitution of functional protein channels into the artificial membranes of levitated ACDC droplets, Remote control of ion channel gating in levitated ACDC droplet core networks)?

Reviewer #2 (Remarks to the Author):

I very much appreciated that the authors accepted and constructively discussed all my suggestions and criticisms to improve their work. In my opinion, the manuscript has now gained clarity and some missed control experiments have also been included to make some results more evident experimentally.

Possible technical improvements to the ACDC platform are also foreseen in the final discussion, although they are not explicitly related to the current limitations of the proposed approach. Hence, this document is ready to be published in its current form.

My only doubt concerns the real novelty compared to the previous articles already published by some of the authors, regarding the impact of the Nature Communication Journal. Since I am not a true expert on microfluidic devices, I may not be able to appreciate the technical novelty enough. Therefore, I leave this issue to the editors and the comments of the other reviewers.

Finally, I would like to thank the authors for their positive approach to reviewing their work.

Reviewer #3 (Remarks to the Author):

The concerns raised in my previous review have been addressed. The general comments from that review still stand and I consider the work worthy of publication.

The paper describes an intriguing application of droplet microfluidics to create synthetic cells followed by in-air acoustic levitation to provide an environment in which the synthetic cells can be manipulated, modified, and, to an extent, controlled. The use of droplet-microfluidics for artificial cell synthesis has received significant attention in recent years but the combination of that approach with acoustic levitation is both novel and ambitious. The droplet patterning, an extension of previously published work from this collaboration, provides significant flexibility in creating controlled, patterned cores for the artificial cells. While use of acoustic manipulation has been reported previously in the generation of synthetic cells, this has been in a liquid continuum while here the in-air levitation allows for contact-free suspension and manipulation of the cells under study, and provides a means of reconfiguring the packing within the droplet.

Overall, this has the potential to provide a valuable platform for creating and studying artificial cells and is worthy of publication. The methodology and interpretation of the observations appear sound and the supplementary information provides useful additional detail that should allow for the approach to be reproduced elsewhere.

Response to Referees second round of review

Reviewer #1 (Remarks to the Author):

I can see that the authors took a great effort to address the review comments, and they did address most of them. However, the main concern as below was not addressed. If the droplets are not stable in the system, it will significantly limit its application where imaging is needed.

“Quite some operations have been achieved in the acoustic levitation, however, the patterning and sizes of the droplets in 3D are in mm-cm scale and seems not quite steady, it will bring significant limitations and challenges for high resolution imaging in biological cell scale (~10 μm). “

It is still not quite clear where the limitation is regarding the size of the droplet that can be handled by the acoustic levitation, with the operations in the manuscript (such as control of ACDC structural organization, microcentrifuge, heating, polarization, Reconstitution of functional protein channels into the artificial membranes of levitated ACDC droplets, Remote control of ion channel gating in levitated ACDC droplet core networks)?

Response: Thank you for pointing out the remaining details of this question. We have added the following additional detail to our first response (above) to evidence the feasibility of application of these or similar processes to smaller structures.

“In previous publications from authors within the collaborative team [1], similar acoustic levitators have been used to trap aerosol spray to “grow” water droplets at the acoustic standing wave nodes in air. Argo and co-worker have demonstrated levitation of aerosol and particulate, which size is down to 1 μm , and further shown that progressively smaller liquid droplets can be captured at correspondingly higher frequencies [2]. The stability of the levitated artificial cells could be enhanced for time-dependent imaging and analytical purposes, using a more stable battery powered source for transducer actuation, or, as we have shown, using an additional ‘twin-trap’ that introduces a stabilising acoustic torque lock for non-spherical particles [3], which technique Kepa and co-workers found to be very stable when studying the protein crystallography of thin films [4] and in snowflake melting studies[5]. Hence, in principle, the presented acoustic levitation and operations can manipulate cell-sized droplets in air, that can be gathered and then controlled and measured collectively with further droplet processing methods and associated imaging equipment development. These limitations and considerations have been addressed in the discussion.”

We have now also added a brief distillation of this to the discussion section of the manuscript and pointed out that approaches such as the magnetic lock approach or other simultaneously applied forces can be used to stabilise the levitated ACDC droplet for the purposes of imaging.

“In addition, aerosol and particulate down to 1 μm can be captured and evaluated using an acoustic levitation method [60]. Meanwhile, the levitated object can be stabilised with shaped acoustic field [61], or approaches comparable to the magnetic positional lock technique employed here for more stabilised imaging. These approaches could also be applicable for higher-resolution imaging of artificial cells like those reported here.”

Reviewer #2 (Remarks to the Author):

I very much appreciated that the authors accepted and constructively discussed all my suggestions and criticisms to improve their work. In my opinion, the manuscript has now gained clarity and some missed control experiments have also been included to make some results more evident experimentally.

Possible technical improvements to the ACDC platform are also foreseen in the final discussion, although they are not explicitly related to the current limitations of the proposed approach.

Hence, this document is ready to be published in its current form.

My only doubt concerns the real novelty compared to the previous articles already published by some of the authors, regarding the impact of the Nature Communication Journal. Since I am not a true expert on microfluidic devices, I may not be able to appreciate the technical novelty enough. Therefore, I leave this issue to the editors and the comments of the other reviewers.

Finally, I would like to thank the authors for their positive approach to reviewing their work.

Reviewer #3 (Remarks to the Author):

The concerns raised in my previous review have been addressed. The general comments from that review still stand and I consider the work worthy of publication.

The paper describes an intriguing application of droplet microfluidics to create synthetic cells followed by in-air acoustic levitation to provide an environment in which the synthetic cells can be manipulated, modified, and, to an extent, controlled. The use of droplet-microfluidics for artificial cell synthesis has received significant attention in recent years but the combination of that approach with acoustic levitation is both novel and ambitious. The droplet patterning, an extension of previously published work from this collaboration, provides significant flexibility in creating controlled, patterned cores for the artificial cells. While use of acoustic manipulation has been reported previously in the generation of synthetic cells, this has been in a liquid continuum while here the in-air levitation allows for contact-free suspension and manipulation of the cells under study, and provides a means of reconfiguring the packing within the droplet.

Overall, this has the potential to provide a valuable platform for creating and studying artificial cells and is worthy of publication. The methodology and interpretation of the observations appear sound and the supplementary information provides useful additional detail that should allow for the approach to be reproduced elsewhere.

Reference:

1. Marzo, A., Barnes, A., & Drinkwater, B. W. TinyLev: A multi-emitter single-axis acoustic levitator. *Review of Scientific Instruments* **88**, 085105 (2017).
2. Argo, T. F., Brian I. V., Zadler, J. and Meegan G. D. Size selection of levitated aerosol particulate in an ultrasonic field. *The Journal of the Acoustical Society of America* **147**, EL93 (2020).
3. Cox, L., Croxford, A., Drinkwater B. W. and Marzo, A. Acoustic Lock: Position and orientation trapping of non-spherical sub-wavelength particles in mid-air using a single-axis acoustic levitator. *Appl. Phys. Lett.* **113**, 054101 (2018).
4. Kepa, M.W. et al. Acoustic levitation and rotation of thin films and their application for room temperature protein crystallography, *Scientific Reports* **12**, 5349 (2022).
5. Köbschall, K. et al. Shape Evolution of a Melting Snowflake. *AIAA Journal*, 2022-3371 (2022).